# ON FEATURE LEARNING IN NEURAL NETWORKS WITH GLOBAL CONVERGENCE GUARANTEES

**Zhengdao Chen**
Courant Institute of Mathematical Sciences
New York University
New York, NY 10012, USA
zc1216@nyu.edu

**Eric Vanden-Eijnden**
Courant Institute of Mathematical Sciences
New York University
New York, NY 10012, USA
eve2@cims.nyu.edu

**Joan Bruna**
Courant Institute of Mathematical Sciences and Center for Data Science
New York University
New York, NY 10012, USA
bruna@cims.nyu.edu

## ABSTRACT

We study the optimization of wide neural networks (NNs) via gradient flow (GF) in setups that allow feature learning while admitting non-asymptotic global convergence guarantees. First, for wide shallow NNs under the mean-field scaling and with a general class of activation functions, we prove that when the input dimension is no less than the size of the training set, the training loss converges to zero at a linear rate under GF. Building upon this analysis, we study a model of wide multi-layer NNs whose second-to-last layer is trained via GF, for which we also prove a linear-rate convergence of the training loss to zero, but *regardless* of the input dimension. We also show empirically that, unlike in the Neural Tangent Kernel (NTK) regime, our multi-layer model exhibits feature learning and can achieve better generalization performance than its NTK counterpart.

## 1 INTRODUCTION

The training of neural networks (NNs) is typically a non-convex optimization problem, but remarkably, simple algorithms like gradient descent (GD) or its variants can usually succeed in finding solutions with low training losses. To understand this phenomenon, a promising idea is to focus on NNs with large widths (a.k.a. under over-parameterization), for which we can derive infinite-width limits under suitable ways to scale the parameters by the widths. For example, under a "$1/\sqrt{width}$" scaling of the weights, the GD dynamics of wide NNs can be approximated by the linearized dynamics around initialization, and as the widths tend to infinity, we obtain the *Neural Tangent Kernel (NTK)* limit of NNs, where the solution obtained by GD coincides with a kernel method [37]. Importantly, theoretical guarantees for optimization and generalization can be obtained for wide NNs under this scaling [19, 5]. Nonetheless, it was pointed out that this NTK analysis replies on a form of *lazy training* that excludes the learning of *features* or *representations* [16, 72], which is a crucial ingredient to the success of deep learning, and is therefore not adequate for explaining the success of NNs [27, 41].

Meanwhile, for shallow (i.e., one-hidden-layer) NNs, if we choose a "*1 / width*" scaling, we can derive an alternative *mean-field (MF)* limit as the widths tend to infinity. Under this scaling, feature learning occurs even in the infinite-width limit, and the training dynamics can be described by the Wasserstein gradient flow of a probability measure on the space of the parameters, which converges to a global minimizer of the loss function under certain conditions [59, 50, 63, 14]. Generalization guarantees have also been proved for learning with shallow NNs under the MF scaling by identifying a corresponding function space [6, 48]. However, currently there are three limitations to this model of over-parameterized NNs. First, the global convergence guarantees for shallow NNs only hold in the infinite-width limit (i.e. they are asymptotic). While [12] studies the deviation between finite-width NNs and their infinite-width limits during training, the analysis is done only asymptotically to the

next order in width. Second, a convergence rate has yet to be established except under special assumptions or with modifications to the GD algorithm [13, 34, 43, 38]. Third, while several works have proposed to extend the MF formulation to deep (i.e., multi-layer) NNs [4, 62, 52, 23, 57], there is less concensus on what the right model should be than for the shallow case. In summary, we still lack a model for the GD optimization of shallow and multi-layer NNs that goes beyond lazy training while admitting fast global convergence.

In this work, we study the optimization of both shallow NNs under the MF scaling and a type of partially-trained multi-layer NNs, and obtain theoretical guarantees of linear-rate global convergence.

## 1.1 SUMMARY OF MAIN CONTRIBUTIONS

We consider the scenario of training NN models to fit a training set of $n$ data points in dimension $d$, where the model parameters are optimized by gradient flow (GF, which is the continuous-time limit of GD) with respect to the squared loss. Allowing most choices of the activation function, we prove that:

1. For a shallow NN, if the hidden layer is sufficiently wide and the input data are linearly independent (requiring $n \leq d$), then with high probability, the training loss converges to zero at a linear rate.

2. For a multi-layer NN where we only train the *second-to-last* layer, if the hidden layers are both sufficiently wide, then with high probability, the training loss converges to zero at a linear rate. Unlike for shallow NNs, here we no longer need the requirement on input dimension, demonstrating a benefit of jointly having depth and width.

We also run numerical experiments to demonstrate that our model exhibits feature learning and can achieve better generalization performance than its NTK counterpart.

## 1.2 RELATED WORKS

**Over-parameterized NNs, NTK and lazy training.** Many recent works have studied the optimization landscape of NNs and the benefits of over-parameterization [24, 67, 60, 66, 64, 54, 65, 42, 3, 40, 76, 15, 75, 17, 39, 9, 32, 20, 31, 61, 10]. One influential idea is the *Neural Tangent Kernel (NTK)* [37], which characterizes the behavior of GD on the infinite-width limit of NNs under a particular scaling of the parameters (e.g. for shallow NNs, replacing $1/m$ with $1/\sqrt{m}$ in (1)). In particular, when the network width is polynomially large in the size of the training set, the training loss converges to a global minimum at a linear rate under GD [19, 5, 55]. Nonetheless, in the NTK limit, due to a relatively large scaling of the parameters at initialization, the hidden-layer features do not move significantly [37, 19, 16]. For this reason, the NTK scaling has been called the *lazy-training* regime, as opposed to a *feature-learning* or *rich* regime [16, 72, 26]. Several works have investigated the differences between the two regimes both in theory [28, 29, 69, 47] and in practice [27, 41]. In addition, several works have generalized the NTK analysis by considering higher-order Taylor approximations of the GD dynamics or finite-width corrections to the NTK [2, 35, 7, 33].

**Mean-field theory of NNs.** An alternative path has been taken to study shallow NNs in the mean-field scaling (as in (1)), where the infinite-width limit is analogous to the thermodynamic or hydrodynamic limit of interacting particle systems [59, 50, 63, 14, 49, 70]. Thanks to the interchangeability of the parameters, the neural network is equivalently characterized by a probability measure on the space of its parameters, and the training can then be described by a Wasserstein gradient flow followed by this probability measure, which, in the infinite-width limit, converges to global mimima under mild conditions. Regarding convergence rate, ref. [71] proves that if we train a shallow NN to fit a Lipschitz target function under population loss, the convergence rate cannot beat the curse of dimensionality. In contrast, we will study the setting of empirical risk minimization, where there are finitely many training data. Ref. [34] shows that mean field Langevin dynamics on shallow NNs can converge exponentially to global minimizers in over-regularized scenarios, but we focus on GF without entropic regularization. Besides the question of optimization, shallow NNs under this scaling represent functions in the Barron space [48] or variation-norm function space [6], which provide theoretical guarantees on generalization as well as fluctuation in training [12]. Several works have proposed different mean-field limits of wide multi-layer NNs and proved convergence

guarantees [4, 62, 51, 52, 23, 57, 22], but questions remain. First, due to the presence of different symmetries in a multi-layer network compared to a shallow network [57], the limiting object at the infinite-width limit is often quite complicated. Second, it has been pointed out that under the MF scaling of a multi-layer network, an i.i.d. initialization of the weights would lead to a collapse of the diversity of neurons in the middle layers, diminishing the effect of having large widths [23]. In addition, while another line of work develops MF models of residual models [8, 46, 21, 36], we are interested in multi-layer NN models with a large width in every layer.

**Feature learning in deep NNs.**  Ref. [1] demonstrates the importance of hierarchical learning by proving the existence of concept classes that can be learned efficiently by a deep NN with quadratic activations but not by non-hierarchical models. Ref. [11] studies the optimization landscape and generalization properties of a hierarchical model that is similar to ours in spirit, where an untrained embedding of the input is passed into a trainable shallow model, and prove an improvement in sample complexity in learning polynomials by having neural network outputs as the embedding. However, the trainable models they consider are not shallow NNs but their linearized and quadratic-Taylor approximations, and furthermore the convergence rate of the training is not known. Ref. [73] proposes a novel parameterization under which there exists an infinite-width limit of deep NNs that exhibits feature learning, but properties of its training dynamics is not well-understood. Our multi-layer NN models adopt an equivalent scaling (see Appendix C), and our focus is on proving non-asymptotic convergence guarantees for its partial training under GF.

## 2  PROBLEM SETUP

### 2.1  MODEL

We summarize our notations in Appendix A. Let $\Omega \subseteq \mathbb{R}^d$ denote the input space, and let $\boldsymbol{x} = [x_1, ..., x_d]^\intercal \in \Omega$ denote a generic input data vector. A shallow NN model under the MF scaling can be written as:

$$f(\boldsymbol{x}) = \frac{1}{m} \sum_{i=1}^{m} c_i \sigma\Big(\frac{1}{\sqrt{d}} \sum_{j=1}^{d} W_{ij} x_j\Big) \,, \tag{1}$$

where $m$ is the width, $W \in \mathbb{R}^{m \times d}$ and $\boldsymbol{c} = [c_1, ..., c_m] \in \mathbb{R}^m$ are the first- and second-layer weight parameters of the model, and $\sigma : \mathbb{R} \to \mathbb{R}$ is the activation function. For simplicity of presentation, we neglect the bias terms. In this paper, we study a more general type of models with the following form:

$$f(\boldsymbol{x}) = \frac{1}{m} \sum_{i=1}^{m} c_i \sigma\big(h_i(\boldsymbol{x})\big) \,, \tag{2}$$

$$\forall i \in [m] \quad : \quad h_i(\boldsymbol{x}) = \frac{1}{\sqrt{D}} \sum_{j=1}^{D} W_{ij} \phi_j(\boldsymbol{x}) \,, \tag{3}$$

where $W \in \mathbb{R}^{m \times D}$ and $\boldsymbol{c} = [c_1, ..., c_m] \in \mathbb{R}^m$ are parameters of the model, and $\phi_1, ..., \phi_D$ are a set of functions from $\Omega$ to $\mathbb{R}$ that we call the *embedding*. Each of $h_1, ..., h_m$ is a function from $\Omega$ to $\mathbb{R}$, and we will refer to them as the *(hidden-layer) feature map* or *activations*. For simplicity, we write $\Phi(\boldsymbol{x}) = [\phi_i(\boldsymbol{x}), ...\phi_D(\boldsymbol{x})]^\intercal \in \mathbb{R}^D$. We consider two types of the embedding, $\Phi$, as described below:

**Fixed embedding**  $D$ is fixed and $\Phi$ is deterministic. In the simplest example, we set $D = d$ and $\phi_j(\boldsymbol{x}) = x_j, \forall j \in [D]$, and recover the shallow NN model in (1). More generally, our definition includes cases where $\Phi$ is a deterministic transformation of an input vector in $\Omega$ into an embedding vector in $\mathbb{R}^D$. This can be understood as input pre-processing or feature engineering.

**High-dimensional random embedding**  $D$ is large and $\Phi$ is random. For instance, we can sample each $\boldsymbol{z}_j$ i.i.d. in $\mathbb{R}^d$ and set $\phi_j(\boldsymbol{x}) = \sigma\Big(\frac{1}{\sqrt{d}} \boldsymbol{z}_j^\intercal \boldsymbol{x}\Big)$, equivalent to setting $\phi_1, ..., \phi_m$ as the hidden-layer

activations of a shallow NN with randomly-initialized first-layer weights. Then, the model becomes

$$f(\boldsymbol{x}) = \frac{1}{m} \sum_{i=1}^{m} c_i \sigma\big(h_i(\boldsymbol{x})\big) \,, \tag{4}$$

$$\forall i \in [m] \quad : \quad h_i(\boldsymbol{x}) = \frac{1}{\sqrt{D}} \sum_{j=1}^{D} W_{ij} \sigma\Big(\frac{1}{\sqrt{d}} \boldsymbol{z}_j^\mathsf{T} \boldsymbol{x}\Big) \,. \tag{5}$$

Thus, we obtain a 3-layer feed-forward NN whose first-layer weights are random and fixed, and we call it a *partially-trained 3-layer (P-3L) NN*. Note that the scaling in this model is different from both the NTK scaling ($1/\sqrt{m}$ instead of $1/m$ in (4)) and the MF scaling for multi-layer NNs adopted in [4, 62, 51, 57, 23] ($1/D$ instead of $1/\sqrt{D}$ in (5)). We show in Appendix B that when $\sigma$ is homogeneous, this scaling is consistent with the Xavier initialization of neural network parameters up to a reparameterization [30, 56]. We also show in Appendix C that in certain cases this scaling is equivalent to the maximum-update parameterization proposed in [73]. Numerical experiments that compare different scalings are described in Section 4.

## 2.2 Training with gradient flow

Consider the scenario of supervised least-squares regression, where we are given a set of $n$ training data points together with their target values, $(\boldsymbol{x}_1, y_1), ..., (\boldsymbol{x}_n, y_n) \in \Omega \times \mathbb{R}$. We fit our models by minimizing the empirical squared loss:

$$\mathcal{L}[f] = \frac{1}{2} \sum_{a=1}^{n} \big(f(\boldsymbol{x}_a) - y_a\big)^2 \,. \tag{6}$$

To do so, we first initialize the parameters randomly by sampling each $c_i$ and $W_{ij}$ i.i.d. from probability measures $\pi_c \in \mathcal{P}(\mathbb{R})$ and $\pi_{\boldsymbol{w}} \in \mathcal{P}(\mathbb{R}^D)$, respectively, and then perform GD on $W$. For simplicity of analysis, we leave **c** *untrained*, and further assume that

**Assumption 1.** $\pi_c = \frac{1}{2}\delta_{\hat{c}}(dc) + \frac{1}{2}\delta_{-\hat{c}}(dc)$ *for some* $\hat{c} > 0$ *independent from m, which is the law of a scaled Rademacher random variable.*

If $\sigma$ is Lipschitz, it is differentiable almost everywhere, and we write $\sigma'(\boldsymbol{x})$ to denote the derivative of $\sigma$ when it is differentiable at $\boldsymbol{x}$ and 0 otherwise. When $\sigma$ is differentiable at $h_i(\boldsymbol{x})$, there is

$$\frac{\partial f}{\partial W_{ij}}(\boldsymbol{x}) = \frac{1}{m\sqrt{D}} c_i \sigma'\big(h_i(\boldsymbol{x})\big) \phi_j(\boldsymbol{x}) \,, \tag{7}$$

and the gradient of the loss function with respect to $W_{ij}$ is given by

$$\frac{\partial \mathcal{L}[f]}{\partial W_{ij}} = \frac{1}{m\sqrt{D}} c_i \sum_{a=1}^{n} \left(f(\boldsymbol{x}_a) - y_a\right) \sigma'\big(h_i(\boldsymbol{x}_a)\big) \phi_j(\boldsymbol{x}_a) \,. \tag{8}$$

Thus, we can perform GD updates on $W$ according to the following rule: $\forall i \in [m]$ and $\forall j \in [D]$,

$$W_{ij} \quad \leftarrow \quad W_{ij} - m\delta\frac{\partial \mathcal{L}[f]}{\partial W_{ij}} = W_{ij} - \frac{\delta}{\sqrt{D}} c_i \sum_{a=1}^{n} \left(f(\boldsymbol{x}_a) - y_a\right) \sigma'\big(h_i(\boldsymbol{x}_a)\big) \phi_j(\boldsymbol{x}_a) \,, \tag{9}$$

where $\delta > 0$ is the step size. As we discuss in Appendix B, this is consistent with the standard GD update rule for Xavier-initialzed NNs. In the limit of infinitesimal step size ($\delta \to 0$), the evolution of the parameters during training is described by the GF equation: if we use the superscript $t \geq 0$ to denote time elapsed during training, the time-derivative of the parameters is given by

$$\dot{W}_{ij}^t = -\frac{c_i}{\sqrt{D}} \sum_{a=1}^{n} \left(f^t(\boldsymbol{x}_a) - y_a\right) \sigma'\big(h_i^t(\boldsymbol{x}_a)\big) \phi_j(\boldsymbol{x}_a) \,, \tag{10}$$

where $f^t$ denotes the output function and $h_1^t, ..., h_m^t$ denote the hidden-layer feature maps determined by the parameters at time $t$. Then, induced by the evolution of $W^t$, each $h_i^t$ evolves according to

$$\forall \boldsymbol{x} \in \Omega \quad : \quad \dot{h}_i^t(\boldsymbol{x}) = \frac{1}{\sqrt{D}} \sum_{j=1}^{D} \dot{W}_{ij}^t \phi_j(\boldsymbol{x}) = -c_i \sum_{a=1}^{n} \mathcal{G}(\boldsymbol{x}, \boldsymbol{x}_a) \left(f^t(\boldsymbol{x}_a) - y_a\right) \sigma'\big(h_i^t(\boldsymbol{x}_a)\big) \,, \tag{11}$$

where we define a kernel function $\mathcal{G} : \Omega \times \Omega \to \mathbb{R}$ as

$$\mathcal{G}(\boldsymbol{x}, \boldsymbol{x}') = \frac{1}{D} \sum_{j=1}^{D} \phi_j(\boldsymbol{x}) \phi_j(\boldsymbol{x}') = \frac{1}{D} (\Phi(\boldsymbol{x}))^{\mathsf{T}} \Phi(\boldsymbol{x}') \, . \tag{12}$$

Accordingly, the output function $f^t$ satisfies

$$\begin{aligned}
\forall \boldsymbol{x} \in \Omega \quad : \quad \dot{f}^t(\boldsymbol{x}) &= \frac{1}{m} \sum_{i=1}^{m} c_i \sigma'\big(h_i^t(\boldsymbol{x})\big) \dot{h}_i^t(\boldsymbol{x}) \\
&= -\frac{\hat{c}^2}{m} \sum_{i=1}^{m} \sigma'\big(h_i^t(\boldsymbol{x})\big) \sum_{a=1}^{n} \mathcal{G}(\boldsymbol{x}, \boldsymbol{x}_a) \left(f^t(\boldsymbol{x}_a) - y_a\right) \sigma'\big(h_i^t(\boldsymbol{x}_a)\big) \, ,
\end{aligned} \tag{13}$$

Thus, the loss function $\mathcal{L}^t := \mathcal{L}[f^t]$ evolves according to

$$\begin{aligned}
\dot{\mathcal{L}}^t &= \sum_{a=1}^{n} \left(f^t(\boldsymbol{x}_a) - y_a\right) \dot{f}^t(\boldsymbol{x}_a) \\
&= -\frac{\hat{c}^2}{m} \sum_{i=1}^{m} \sum_{a,b=1}^{n} G_{ab} \left(f^t(\boldsymbol{x}_a) - y_a\right) \left(f^t(\boldsymbol{x}_b) - y_b\right) \sigma'\big(h_i(\boldsymbol{x}_a)\big) \sigma'\big(h_i(\boldsymbol{x}_b)\big) \\
&\leq -\frac{\hat{c}^2 \lambda_{\min}(G)}{m} \sum_{i=1}^{m} \sum_{a=1}^{n} \left(f^t(\boldsymbol{x}_a) - y_a\right)^2 \left(\sigma'\big(h_i(\boldsymbol{x}_a)\big)\right)^2 \, ,
\end{aligned} \tag{14}$$

where we define the (symmetric) *Gram matrix* $G \in \mathbb{R}^{n \times n}$ with entries $G_{ab} = G_{ba} = \mathcal{G}(\boldsymbol{x}_a, \boldsymbol{x}_b)$, and use $\lambda_{\min}(G)$ to denote its least eigenvalue. We will also use $G_{\min} = \min_{a \in [n]} G_{aa}$ and $G_{\max} = \max_{a \in [n]} G_{aa}$ to denote the minimum and maximum diagonal entries of $G$, respectively. Since $G$ is positive semi-definite, we see that $\dot{\mathcal{L}}^t \leq 0$, which means that the loss value is indeed non-increasing along the GF trajectory.

**Feature learning**  Compared to the NTK scaling of neural networks, the crucial difference is the $1/m$ factor in (2), instead of $1/\sqrt{m}$. It is known that under the NTK scaling, due to the $1/\sqrt{m}$ factor, the movement of the feature maps, $h_1, ..., h_m$, is only of order $O(1/\sqrt{m})$ while the function value changes by an amount of order $\Omega(1)$. While this greatly simplifies the convergence analysis, it also implies that the hidden-layer representations are not being learned. In contrast, with the $1/m$ factor in (2), if $\sigma$ is Lipschitz with Lipschitz constant $L_\sigma$, there is $|f^{t_2}(\boldsymbol{x}) - f^{t_1}(\boldsymbol{x})| \leq \frac{L_\sigma \hat{c}}{m} \sum_{i=1}^{m} |h_i^{t_2}(\boldsymbol{x}) - h_i^{t_1}(\boldsymbol{x})|, \forall t_1, t_2 \geq 0$. Therefore, regardless of $m$ and $D$,

$$\frac{1}{m} \sum_{i=1}^{m} |h_i^{t_1}(\boldsymbol{x}) - h_i^{t_2}(\boldsymbol{x})| \geq (\hat{c})^{-1} (L_\sigma)^{-1} |f^{t_1}(\boldsymbol{x}) - f^{t_2}(\boldsymbol{x})| \, , \tag{15}$$

which implies that the average movement of the feature maps is on the same order as the change in function value, and thus the hidden-layer representations as well as the NTK undergoes nontrivial movement during training. In Appendix C, we further justify the occurrence of feature learning using the framework developed in [73].

## 3  CONVERGENCE ANALYSIS

### 3.1  MODELS WITH A FIXED EMBEDDING

To prove that the training loss converges to zero, we need a lower bound on the absolute value of $\dot{\mathcal{L}}_t$. Indeed, if $G$ is positive definite, which depends on $\Phi$ and the training data, we can establish one in the following way. First, as a simple case, if we use an activation function whose derivative's absolute value is uniformly bounded from below by a constant $K_{\sigma'} > 0$, such as linear, cubic or (smoothed) Leaky ReLU activations, we can derive a Polyak-Lojasiewicz (PL) condition [58, 45] from (14) directly,

$$\dot{\mathcal{L}}^t \leq -2\hat{c}^2 \lambda_{\min}(G) \left(K_{\sigma'}\right)^2 \mathcal{L}^t \, , \tag{16}$$

which implies $\mathcal{L}_t \leq \mathcal{L}_0 e^{-2\hat{c}^2 \lambda_{\min}(G)(K_{\sigma'})^2 t}$, indicating that the training loss decays to $0$ at a linear rate.

For more general choices of the activation function, a challenge is to guarantee that, heuristically speaking, for each $a \in [n]$, $\sigma'\big(h_i(\boldsymbol{x}_a)\big)$ does not become near zero for too many $i \in [m]$ before the loss vanishes. To facilitate a finer-grained analysis, we need the following mild assumption on $\sigma$:

**Assumption 2.** $\sigma$ *is Lipschitz with Lipschitz constant* $L_\sigma$, *and there exists an open interval* $I = (I_l, I_r) \subseteq \mathbb{R}$ *on which* $\sigma$ *is differentiable and* $|\sigma'|$ *is lower-bounded by some* $K_{\sigma'} > 0$.

Intuitively, $I$ is an *active region* of $\sigma$, within which the derivative has a magnitude bounded away from zero. This assumption is satisfied by the majority of activation functions in practice, including smooth ones such as $\tanh$ and sigmoid as well as non-smooth ones such as ReLU. Then, under the following initialization scheme, we prove a general result for models with a fixed embedding.

**Assumption 3.** $\pi_{\boldsymbol{w}}$ *is the $D$-dimensional standard Gaussian distribution, i.e., each $W_{ij}$ is sampled independently from a standard Gaussian distribution.*

**Theorem 1** (Fixed embedding). *Suppose that Assumptions 1, 2 and 3 are satisfied, and $\lambda_{\min}(G) > 0$. Then $\exists \hat{c}_0$, $r$ and $C > 0$ such that $\forall \delta > 0$, if $\hat{c} \geq \hat{c}_0 \lambda_{\max}(G)/\lambda_{\min}(G)$ and $m \geq C(1 + \hat{c}^2) \log(n/\delta)$, then with probability at least $1 - \delta$, it holds that $\forall t \geq 0$,*

$$\mathcal{L}_t \leq \mathcal{L}_0 e^{-r\hat{c}^2 \lambda_{\min}(G)t} . \tag{17}$$

*Here, $\hat{c}_0$, $r$ and $C$ depend on $I$, $G_{\min}$, $G_{\max}$, $\|\boldsymbol{y}\|$, $L_\sigma$ and $K_{\sigma'}$ (but not on $m$, $n$, $d$, $D$, $\delta$, or $\lambda_{\min}(G)$).*

The result is proved in Appendix E, and below we briefly describe the intuition. A key to the proof is to guarantee that enough neurons remain in the active region throughout training. Specifically, with respect to each training data point (i.e. for each $a \in [n]$), we can keep track of the proportion of neurons (among all $i \in [m]$) for which $h_i^t(\boldsymbol{x}_a) \in I$. We show that if the proportion is large enough at initialization (shown by Lemma 3 in Appendix E.2 under Assumption 3), then it cannot drop dramatically without a simultaneous decrease of the loss value, as long as the $c_i$'s are not too small in absolute value. This property of the dynamics is formalized in the following lemma:

**Lemma 1.** *Consider the dynamics of $\mathcal{L}^t$ and $\big\{h_i^t(\boldsymbol{x}_a)\big\}_{i \in [m], a \in [n]}$ governed by (11) and (14). Assume that $\lambda_{\min}(G) > 0$, and $\forall i \in [m]$, $|c_i| = \hat{c} > 0$. Under Assumption 2, define*

$$\eta^t = \min_{a \in [n]} \left\{ \frac{1}{m} \sum_{i=1}^{m} \mathbb{1}_{h_i^t(\boldsymbol{x}_a) \in I} \right\} , \forall t \geq 0 \quad and \quad \tilde{\eta}^0 = \min_{a \in [n]} \left\{ \frac{1}{m} \sum_{i=1}^{m} \mathbb{1}_{h_i^0(\boldsymbol{x}_a) \in (\frac{2I_l + I_r}{3}, \frac{I_l + 2I_r}{3})} \right\} \tag{18}$$

*Then $\forall t \geq 0$, there is*

$$\big(\eta^t\big)^{\frac{3}{2}} \geq \big(\tilde{\eta}^0\big)^{\frac{3}{2}} - \kappa\big(\lambda_{\min}(G), \lambda_{\max}(G)\big)\big((\mathcal{L}^0)^{\frac{1}{2}} - (\mathcal{L}^t)^{\frac{1}{2}}\big)/\hat{c} , \tag{19}$$

*where $\kappa(\lambda_1, \lambda_2) = \frac{9\lambda_2(I_r - I_l)}{2\lambda_1 K_{\sigma'}}$.*

### 3.1.1 EXAMPLE: SHALLOW NEURAL NETWORKS WHEN $n \leq d$

In the case of shallow NNs under the MF scaling, $G = G^{(0)} \in \mathbb{R}^{n \times n}$, where

$$G_{ab}^{(0)} := \frac{1}{d}(\boldsymbol{x}_a)^\intercal \boldsymbol{x}_b \tag{20}$$

Thus, $G^{(0)}$ is positive definite if and only the training data set $\{\boldsymbol{x}_1, ..., \boldsymbol{x}_n\} \subseteq \mathbb{R}^d$ consists of linearly-independent vectors, which is possible (and expected if the training data are sampled independently from some non-degenerate distribution) when $n \leq d$. In that case, Theorem 1 implies

**Corollary 1** (Shallow NN with $n \leq d$). *Suppose that Assumptions 1, 2 and 3 are satisfied. If the training data are linearly-independent vectors, then under GF (10) on the first-layer weights of the shallow NN, the training loss converges to zero at a linear rate.*

While the assumption that $n \leq d$ is restrictive, we note that existing convergence rate guarantees for the GD-type training of shallow NNs in the MF scaling need strong additional assumptions [38], modifications to the GD algorithm [34, 13, 53], or restrictions to certain special tasks [43].

## 3.2 MODELS WITH A HIGH-DIMENSIONAL RANDOM EMBEDDING

A clear limitation of Corollary 1 is that it is only applicable when $n \leq d$, since otherwise the Gram matrix $G^{(0)}$ cannot be positive definite. This motivates us to consider the use of a high-dimensional embedding $\Phi$ to lift the effective input dimension. In particular, we focus on the scenario where $D$ is large and $\Phi$ is random. While the Gram matrix $G$ in this case is also random, we only need that it concentrates around a deterministic and positive definite limit as $D$ tends to infinity:

**Condition 1** (Concentration of $G$ around a positive definite matrix). *There exists a (deterministic) positive definite matrix $\bar{G} \in \mathbb{R}^{n \times n}$ with least eigenvalue $\lambda_{\min}(\bar{G}) > 0$ such that $\forall \delta, u > 0$, $\exists D_{\min}(\delta, u) > 0$ such that if $D \geq D_{\min}(\delta, u)$, then $\mathbb{P}(\|G - \bar{G}\|_2 > u) < \delta$.*

Condition 1 is sufficient for us to apply Lemma 1 and obtain the following global convergence guarantee, which extends Theorem 1 to models with a high-dimensional random embedding. The proof is given in Appendix F.

**Theorem 2** (High-dimensional random embedding). *Under Assumptions 1, 2, 3 and Condition 1, $\exists \hat{c}_0$, $r$ and $C > 0$ such that $\forall \delta > 0$, if $\hat{c} \geq \hat{c}_0 \lambda_{\max}(\bar{G})/\lambda_{\min}(\bar{G})$, $m \geq C(1 + \hat{c}^2) \log(n/\delta)$ and $D \geq D_{\min}(\frac{1}{2}\delta, \frac{1}{2}\lambda_{\min}(\bar{G}))$, then with probability at least $1 - \delta$, it holds that $\forall t \geq 0$,*

$$\mathcal{L}_t \leq \mathcal{L}_0 e^{-r\hat{c}^2 \lambda_{\min}(G)t} . \tag{21}$$

*Here, $\hat{c}_0$, $r$ and $C$ depend on $I, \bar{G}_{\min}, \bar{G}_{\max}, \|\boldsymbol{y}\|, L_\sigma$ and $K_{\sigma'}$ (but not $m$, $n$, $d$, $D$, $\delta$, or $\lambda_{\min}(\bar{G})$).*

### 3.2.1 EXAMPLE: PARTIALLY-TRAINED THREE-LAYER NEURAL NETWORKS (P-3L NNS)

Consider the P-3L NN model defined in (4). In this case, the Gram matrix is $G^{(1)}$, defined by

$$G^{(1)}_{ab} = \frac{1}{D} \sum_{j=1}^{D} \sigma\Big(\frac{1}{\sqrt{d}} \boldsymbol{z}_j^\mathsf{T} \boldsymbol{x}_a\Big) \sigma\Big(\frac{1}{\sqrt{d}} \boldsymbol{z}_j^\mathsf{T} \boldsymbol{x}_b\Big) \tag{22}$$

If $\boldsymbol{z}_1, ..., \boldsymbol{z}_m$ are sampled i.i.d. from a probability measure $\pi_{\boldsymbol{z}}$ on $\mathbb{R}^d$ and fixed during training, then the limiting Gram matrix, denoted by $\bar{G}^{(1)} \in \mathbb{R}^{n \times n}$, is given by

$$\bar{G}^{(1)}_{ab} = \mathbb{E}_{\boldsymbol{z} \sim \pi_{\boldsymbol{z}}} \left[ \sigma\Big(\frac{1}{\sqrt{d}} \boldsymbol{z}^\mathsf{T} \boldsymbol{x}_a\Big) \sigma\Big(\frac{1}{\sqrt{d}} \boldsymbol{z}^\mathsf{T} \boldsymbol{x}_b\Big) \right] \tag{23}$$

Thus, for the convergence result, the assumption we need on the limiting Gram matrix is

**Assumption 4.** *$\pi_{\boldsymbol{z}}$ is sub-Gaussian and the matrix $\bar{G}^{(1)}$, which depends on the choice of $\sigma$ and the training set, is positive definite with $\lambda_{\min}(\bar{G}^{(1)}) > 0$ and $(\bar{G}^{(1)})_{\max} < \infty$.*

This assumption also plays an important role in the NTK analysis, and it is satisfied if, for example, $\pi_{\boldsymbol{z}}$ is the $d$-dimensional standard Gaussian distribution, no two data points are parallel, and $\sigma$ is either the ReLU function [19] or analytic and not a polynomial [18]. When Assumption 4 is satisfied, as long as $\sigma$ is Lipschitz, we can use standard concentration techniques to verify Condition 1. Thus, Theorem 2 implies that

**Theorem 3** (P-3L NN). *Under Assumptions 1, 2, 3 and 4, $\exists \hat{c}_0$, $r$, $C_1$ and $C_2 > 0$ such that $\forall \delta > 0$, if $\hat{c} \geq \hat{c}_0 \lambda_{\max}(\bar{G}^{(1)})/\lambda_{\min}(\bar{G}^{(1)})$, $m \geq C_1(1 + \hat{c}^2) \log(n/\delta)$ and $D \geq C_2 n^2 \log(n/\delta)/\lambda_{\min}(\bar{G}^{(1)})^2$, then with probability at least $1 - \delta$, it holds that $\forall t \geq 0$,*

$$\mathcal{L}_t \leq \mathcal{L}_0 e^{-r\hat{c}^2 \lambda_{\min}(\bar{G}^{(1)})t} . \tag{24}$$

*Here, $\hat{c}_0$, $r$, $C_1$ and $C_2$ depend on $I, \bar{G}^{(1)}_{\min}, \bar{G}^{(1)}_{\max}, \|\boldsymbol{y}\|, K_{\sigma'}$ as well as the sub-Gaussian norm of $\mu_{\boldsymbol{z}}$ (but not on $m$, $n$, $d$, $D$, $\delta$ or $\lambda_{\min}(\bar{G}^{(1)})$).*

The proof is given in Appendix G. Compared to Corollary 1 for shallow NNs, a highlight of Theorem 3 is that the requirement of $n \leq d$ is no longer needed. This demonstrates an advantage of the high-dimensional random embedding realized by the first hidden layer in the P-3L NN, thus illustrating a benefit of having both depth and width in NNs from the viewpoint of optimization. Compared to the NTK result [18], our analysis assumes the same level of over-parameterization, but crucially allows feature training to occur, which we discuss in Section 2.2 and support empirically in Section 4.3.

Furthermore, by using a *multi-layer* NN with random and fixed weights as the high-dimensional random embedding, we extend the P-3L NN to a *partially-trained L-layer NN* model in Appendix H, for which similar convergence results can be proved for training its second-to-last layer via GF.

Table 1: Three different scalings of the partially-trained 3L NN model considered in Experiment 3.

| Model | Ours | NTK | MF [4, 52, 62, 23] |
|---|---|---|---|
| $f(\boldsymbol{x})$ | $\frac{1}{m}\sum_{i=1}^{m} c_i \sigma\big(h_i(\boldsymbol{x})\big)$ | $\frac{1}{\sqrt{m}}\sum_{i=1}^{m} c_i \sigma\big(h_i(\boldsymbol{x})\big)$ | $\frac{1}{m}\sum_{i=1}^{m} c_i \sigma\big(h_i(\boldsymbol{x})\big)$ |
| $h_i(\boldsymbol{x})$ | $\frac{1}{\sqrt{m}}\sum_{j=1}^{m} W_{ij}\sigma(\frac{1}{\sqrt{d}}\boldsymbol{z}_j^\mathsf{T}\boldsymbol{x})$ | $\frac{1}{\sqrt{m}}\sum_{j=1}^{m} W_{ij}\sigma(\frac{1}{\sqrt{d}}\boldsymbol{z}_j^\mathsf{T}\boldsymbol{x})$ | $\frac{1}{m}\sum_{j=1}^{m} W_{ij}\sigma(\frac{1}{\sqrt{d}}\boldsymbol{z}_j^\mathsf{T}\boldsymbol{x})$ |
| $W_{ij}^{k+1}$ | $W_{ij}^k - m\delta\frac{\partial \mathcal{L}^k}{\partial W_{ij}^k}$ | $W_{ij}^k - \delta\frac{\partial \mathcal{L}^k}{\partial W_{ij}^k}$ | $W_{ij}^k - m^2\delta\frac{\partial \mathcal{L}^k}{\partial W_{ij}^k}$ |

## 4 NUMERICAL EXPERIMENTS

Additional results and details of the experiments are provided in Appendix I.

### 4.1 EXPERIMENT 1: CONVERGENCE OF TRAINING WHEN FITTING RANDOM DATA

We train shallow NNs to fit a randomly labeled data set $\{(\boldsymbol{x}_1, y_1), ..., (\boldsymbol{x}_n, y_n)\}$ with $d = 20$. Specifically, we sample each $\boldsymbol{x}_a$ i.i.d. with every entry sampled independently from a standard Gaussian distribution, and each $y_a$ i.i.d. uniformly on $[-\frac{1}{2}, \frac{1}{2}]$ and independently from the $\boldsymbol{x}_a$'s. We see from Figure 1 that the convergence happens at a nearly linear rate when $n = 20$ and $40$, and the rate decreases as $n$ becomes larger. This is coherent with our theoretical result (Corollary 1), and interestingly also echoes a prior result that the convergence rate of optimizing a shallow NN using *population* loss can suffer from the curse of dimensionality [71], which implies a worsening of the convergence rate as the number of data points increases.

### 4.2 EXPERIMENT 2: BENEFIT OF INPUT EMBEDDING

We consider a model defined by (2) and (3) with $d = 30$ and $\Phi(\boldsymbol{x}) = \text{vec}(\boldsymbol{x}\boldsymbol{x}^\mathsf{T}) \in \mathbb{R}^{d^2}$, which we call a shallow NN augmented with *quadratic embedding*. We compare this model against a plain shallow NN (without the extra embedding), both with $m = 8192$, to fit a series of training sets with various sizes where the target $y$ is given by another shallow NN augmented with quadratic embedding with $m = 5$. We see from Figure 2 that the augmented shallow NN achieves lower test error given the same number of training samples, demonstrating the benefit of a good embedding.

### 4.3 EXPERIMENT 3: FEATURE LEARNING V.S. LAZY TRAINING

We consider the P-3L NN model defined in (4) and (5) with $D = m$ (i.e. both hidden layers having the same width), and compare it with 3-layer NN models under NTK and MF scalings, as we define in Table 1 based on prior literature [37, 4, 52, 62, 23], which undergo partial training in the same fashion. We adopt the data set used in [69] (more details in Appendix I.3), and train the models by minimizing the unregularized squared loss for varying $n$'s and $m$'s.

First, we see from the top-left plot in Figure 4 that, consistently across different $m$, the training loss converges at a linear rate for the model under our scaling, which is coherent with Theorem 3. Second, we see from the second row that feature learning occurs in the model under our scaling but negligibly in the model under the NTK scaling, as expected [16]. Note also that under the MF scaling, the feature maps $h_1(\boldsymbol{x}), ..., h_m(\boldsymbol{x})$ concentrate near 0 at initialization due to the small scaling, but gains diversity during training. Third, we see from Figure 3 that our model yields the smallest test errors out of all three, and in addition, as $n$ grows the test error decreases faster under the MF scaling than under the NTK scaling, both indicating an advantage of feature learning compared to lazy training.

## 5 CONCLUSIONS AND LIMITATIONS

We consider a general type of models that includes shallow and partially-trained multi-layer NNs, which exhibits feature learning when trained via GF, and prove non-asymptotic global convergence guarantees that accommodates a general class of activation functions. For a randomly-initialized shallow NN in the MF scaling that is wide enough, we prove that by performing GF on the input-layer weights, the training loss converges to zero at a linear rate if the number of training data does not exceed the input dimension. For a randomly-initialized multi-layer NN with large widths, we prove

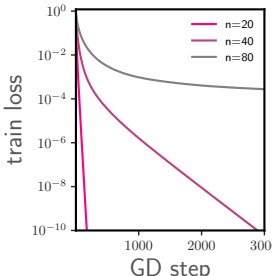
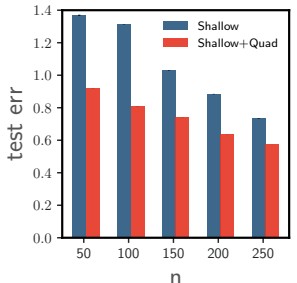
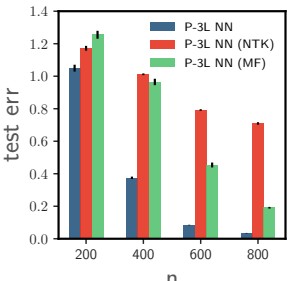

Figure 1: Training loss v.s. number of GD steps for different $n$ in Experiment 1.

Figure 2: Test error v.s. $n$ in Experiment 2 by the two models with $m = 8192$.

Figure 3: Test error v.s. $n$ in Experiment 3 by the three models with $m = 8192$.

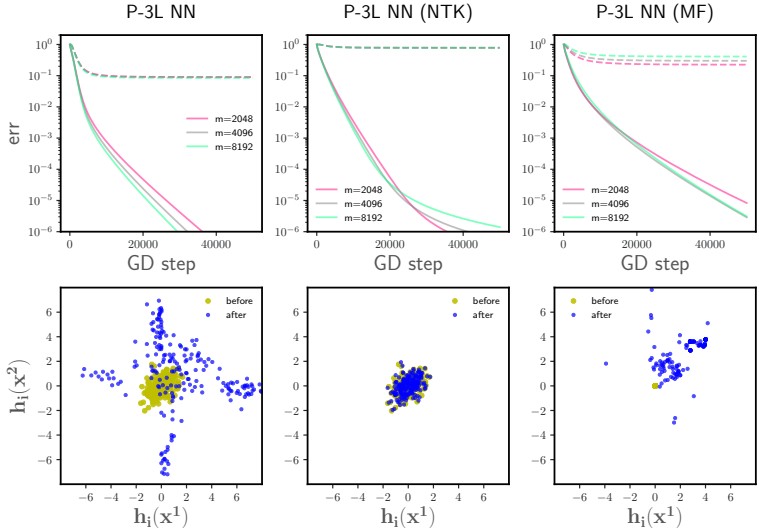

Figure 4: Results of Experiment 3 when $n = 600$. Each column corresponds to a different scaling of the P-3L NN model, as described in Table 1. *Row 1*: Evolution of training loss (solid curve) and test error (dashed curve) during training. *Row 2*: Distribution of the hidden-layer feature map (pre-activation) associated with two particular input data points. Each dot represents a different $i$, (i.e., neuron in the second hidden layer,) and the $x$- and $y$-coordinates equal $h_i(\boldsymbol{x}_1)$ and $h_i(\boldsymbol{x}_2)$, respectively, where $\boldsymbol{x}_1$ is an input from the training set and $\boldsymbol{x}_2$ is an input from the test set.

that by performing GF on the weights in the second-to-last layer, the same result holds except there is no requirement on the input dimension. We also perform numerical experiments to demonstrate the advantage of feature learning in our partially-trained multi-layer NNs relative to their counterparts under the NTK scaling.

Our work focuses on the optimization rather than the approximation or generalization properties of NNs, which are also crucial to understand. In addition, as our current theoretical results on global convergence neglect the bias terms and assume that the last-layer weights are untrained, a more general version is left for future work.

## ACKNOWLEDGMENTS

The authors acknowledge support from the Henry MacCracken Fellowship, NSF RI-1816753, NSF CAREER CIF 1845360, NSF CHS-1901091 and NSF DMS-MoDL 2134216.

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

# A    ADDITIONAL NOTATIONS

- For a positive integer $n$, we let $[n]$ denote the set $\{1, ..., n\}$.
- We use $i, j$ (as subscripts) to index the neurons in the hidden layers, $a, b$ (as subscripts or superscripts) to index different training data points, $t$ (as a superscript) to denote the training time / time parameter in gradient flow.
- We write $\overline{\sum_a}$ for $\frac{1}{n} \sum_{a=1}^{n}$.
- We use bold letters (e.g. $\boldsymbol{x}$, $\boldsymbol{z}$, $\boldsymbol{c}$, $\boldsymbol{y}$) to denote vectors.
- We use $W$ and $\{W_{ij}\}_{i \in [m], j \in [D]}$ interchangeably to refer to the same set of parameters.

# B    CONSISTENCY OF THE SCALING AND GD UPDATE RULE WITH XAVIER INITIALIZATION

Consider a three-layer network defined by

$$f(\boldsymbol{x}) = \sum_{i=1}^{m} \theta_i^{(3)} \sigma\big(h_i(\boldsymbol{x})\big) \tag{25}$$

$$\forall i \in [m] \quad : \quad h_i(\boldsymbol{x}) = \sum_{j=1}^{m} \theta_{ij}^{(2)} \sigma\Big(\frac{1}{\sqrt{d}} \sum_{k=1}^{d} \theta_{jk}^{(1)} x_k\Big) \tag{26}$$

with weight parameters $\{\theta_{jk}^{(1)}\}_{j,k \in [m]}$, $\{\theta_{ij}^{(2)}\}_{i,j \in [m]}$ and $\{\theta_i^{(3)}\}_{i \in [m]}$ are initialized according to Xavier initialization, which means that we sample each $\theta_{jk}^{(1)}$ i.i.d. from $\mathcal{N}(0, \frac{1}{m+d})$, each $\theta_{ij}^{(2)}$ i.i.d. from $\mathcal{N}(0, \frac{1}{2m})$, and each $\theta_i^{(3)}$ i.i.d. from $\mathcal{N}(0, \frac{1}{m+1})$. If $m \gg d$, both $\mathcal{N}(0, \frac{1}{m+d})$ and $\mathcal{N}(0, \frac{1}{m+1})$ can be approximated by $\mathcal{N}(0, \frac{1}{m})$. Then, up to this approximation, by redefining $c_i = \sqrt{m}\theta_i^{(3)}$, $W_{ij} = \sqrt{m}\theta_{ij}^{(2)}$ and $z_{jk} = \sqrt{m}\theta_{jk}^{(1)}$, we can write

$$f(\boldsymbol{x}) = \frac{1}{\sqrt{m}} \sum_{i=1}^{m} c_i \sigma\big(h_i(\boldsymbol{x})\big) , \tag{27}$$

$$\forall i \in [m] \quad : \quad h_i(\boldsymbol{x}) = \frac{1}{\sqrt{m}} \sum_{j=1}^{D} W_{ij} \sigma\Big(\frac{1}{\sqrt{md}} \boldsymbol{z}_j^\intercal \boldsymbol{x}\Big) , \tag{28}$$

and note that $c_i, W_{ij}$ and $z_{jk}$ are all initialized i.i.d. of order $O(1)$. In addition, if $\sigma$ is homogeneous, this is then equivalent to (4) and (5) when $D = m$.

Moreover, there is $\frac{\partial f}{\partial W_{ij}}(\boldsymbol{x}) = \frac{1}{\sqrt{m}} \frac{\partial f}{\partial \theta_{ij}^{(2)}}(\boldsymbol{x})$. Then, since performing GD on $\theta_{ij}^{(2)}$ with step size $\delta$ means updating $\theta_{ij}^{(2)}$ according to

$$\theta_{ij}^{(2)} \leftarrow \theta_{ij}^{(2)} - \delta \frac{\partial \mathcal{L}[f]}{\partial \theta_{ij}^{(2)}} , \tag{29}$$

this is equivalent to updating $W_{ij}$ according to

$$\begin{aligned} W_{ij} \leftarrow &\sqrt{m}\big(\theta_{ij}^{(2)} - \delta \frac{\partial \mathcal{L}[f]}{\partial \theta_{ij}^{(2)}}\big) \\ =& W_{ij} - m\delta \frac{\partial \mathcal{L}[f]}{\partial W_{ij}}(\boldsymbol{x}) , \end{aligned} \tag{30}$$

which justifies the $m$ factor on the right-hand-side of (9).

## C   RELATIONSHIP TO THE MAXIMUM-UPDATE PARAMETERIZATION AND FEATURE LEARNING

Consider the partially-trained $L$-layer NN model defined in Section H in the case where $D = m \gg d$. In the framework of abc-parameterization introduced in [73], our model corresponds to setting

$$a_1 = 0, \ a_2 = ... = a_L = \frac{1}{2}, \ a_{L+1} = 1$$
$$b_l = 0, \ \forall l \in [L+1]$$

Furthermore, as we explain in Appendix B, the appropriate learning rate scales linearly with $m$ (as in (9)), which corresponds to having

$$c = -1 \tag{31}$$

Meanwhile, the maximum-update ($\mu$P) parameterization [73] is characterized by setting

$$a_1 = -\frac{1}{2}, \ a_2 = ... = a_L = 0, \ a_{L+1} = \frac{1}{2} \tag{32}$$

$$b_l = \frac{1}{2}, \ \forall l \in [L+1] \tag{33}$$

$$c = 0 \tag{34}$$

Recall the symmetry of abc-parameterization derived in [73], which states that one gets a different but equivalent abc-parameterization by setting

$$a_l \leftarrow a_l + \theta, \ b_l \leftarrow b_l - \theta, \ c \leftarrow c - 2\theta \tag{35}$$

Since our parameterization can be obtained from the maximum-update parameterization by applying the transformation above with $\theta = \frac{1}{2}$, they are equivalent in the function space. In particular, for our parameterization, the $r$ parameter defined in [73] can be computed as

$$
\begin{aligned}
r &= \min\{b_{L+1}, a_{L+1} + c\} + a_{L+1} + c + \min_{l=1,...,L}\{2a_l - \mathbb{1}_{l \neq 1}\} \\
&= \min\{0, 1 + (-1)\} + 1 + (-1) + \min\{2 \cdot 0 - 0, 2 \cdot \frac{1}{2} - 1\} \\
&= 0
\end{aligned}
\tag{36}
$$

Hence, according to [73], our parameterization exhibits feature learning.

## D   PROOF OF LEMMA 1

Since we assume that $G$ is positive definite and $|c_i| = \hat{c} > 0, \ \forall i \in [m]$, we can derive from (14) that

$$
\begin{aligned}
\dot{\mathcal{L}}^t &= -\frac{\hat{c}^2}{m} \sum_{i=1}^{m} \sum_{a,b=1}^{n} \left(f^t(\boldsymbol{x}_a) - y_a\right)\left(f^t(\boldsymbol{x}_b) - y_b\right) \sigma'\left(h_i^t(\boldsymbol{x}_a)\right)\sigma'\left(h_i^t(\boldsymbol{x}_a)\right) G_{ab} \\
&\leq -\hat{c}^2 \lambda_{\min}(G) \frac{1}{m} \sum_{i=1}^{m} \sum_{a=1}^{n} \left(f^t(\boldsymbol{x}_a) - y_a\right)^2 \left(\sigma'\left(h_i(\boldsymbol{x}_a)\right)\right)^2 \\
&= -\hat{c}^2 \lambda_{\min}(G) \sum_{a=1}^{n} \left(f^t(\boldsymbol{x}_a) - y_a\right)^2 \frac{1}{m} \sum_{i=1}^{m} \left(\sigma'\left(h_i(\boldsymbol{x}_a)\right)\right)^2 \\
&\leq -\hat{c}^2 \lambda_{\min}(G) \sum_{a=1}^{n} \left(f^t(\boldsymbol{x}_a) - y_a\right)^2 \frac{1}{m} \sum_{i=1}^{m} \mathbb{1}_{h_i^t(\boldsymbol{x}_a) \in I}\left(K_{\sigma'}\right)^2 \\
&\leq -\hat{c}^2 \lambda_{\min}(G)\left(K_{\sigma'}\right)^2 \sum_{a=1}^{n} \left(f^t(\boldsymbol{x}_a) - y_a\right)^2 \min_{b \in [n]}\left\{\frac{1}{m} \sum_{i=1}^{m} \mathbb{1}_{h_i^t(\boldsymbol{x}_b) \in I}\right\} \\
&\leq -2\hat{c}^2 \lambda_{\min}(G)\left(K_{\sigma'}\right)^2 \mathcal{L}^t \min_{a \in [n]}\left\{\frac{1}{m} \sum_{i=1}^{m} \mathbb{1}_{h_i^t(\boldsymbol{x}_a) \in I}\right\}
\end{aligned}
\tag{37}
$$

Since $I$ is an open interval, $\exists \xi > 0$ such that we can find a subinterval $I_0 \subseteq I$ such that the distance between $I_0$ and the boundaries of $I$ (if $I$ is bounded on either side) is no less than $\xi$, i.e.,

$$\inf_{u \in I_0, u' \in \mathbb{R} \setminus I} |u - u'| \geq \xi \tag{38}$$

In particular, we can choose $\xi = \frac{1}{3}(I_r - I_l)$ and $I_0 = (I_l + \xi, I_r - \xi)$. Then there is

$$\begin{aligned}
\mathbb{1}_{h_i^t(\boldsymbol{x}_a) \in I} &\geq \mathbb{1}_{h_i^0(\boldsymbol{x}_a) \in I_0 \,,\, h_i^t(\boldsymbol{x}_a) \in I} \\
&\geq \mathbb{1}_{h_i^0(\boldsymbol{x}_a) \in I_0 \,,\, |h_i^t(\boldsymbol{x}_a) - h_i^0(\boldsymbol{x}_a)| < \xi} \\
&\geq \mathbb{1}_{h_i^0(\boldsymbol{x}_a) \in I_0} - \mathbb{1}_{|h_i^t(\boldsymbol{x}_a) - h_i^0(\boldsymbol{x}_a)| \geq \xi}
\end{aligned} \tag{39}$$

and so

$$\min_{a \in [n]} \left\{ \frac{1}{m} \sum_{i=1}^m \mathbb{1}_{h_i^t(\boldsymbol{x}_a) \in I} \right\} \geq \min_{a \in [n]} \left\{ \frac{1}{m} \sum_{i=1}^m \mathbb{1}_{h_i^0(\boldsymbol{x}_a) \in I_0} \right\} - \max_{a \in [n]} \left\{ \frac{1}{m} \sum_{i=1}^m \mathbb{1}_{|h_i^t(\boldsymbol{x}_a) - h_i^0(\boldsymbol{x}_a)| \geq \xi} \right\} \tag{40}$$

Thus, we have

$$\dot{\mathcal{L}}^t \leq -2\hat{c}^2 \lambda_{\min}(G) \left(K_{\sigma'}\right)^2 \mathcal{L}^t \left( \min_{a \in [n]} \left\{ \frac{1}{m} \sum_{i=1}^m \mathbb{1}_{h_i^0(\boldsymbol{x}_a) \in I_0} \right\} - \max_{a \in [n]} \left\{ \frac{1}{m} \sum_{i=1}^m \mathbb{1}_{|h_i^t(\boldsymbol{x}_a) - h_i^0(\boldsymbol{x}_a)| \geq \xi} \right\} \right) \tag{41}$$

Meanwhile, since

$$\dot{h}_i^t(\boldsymbol{x}_a) = -c_i \sum_{b=1}^n \left( f^t(\boldsymbol{x}_b) - y_b \right) \sigma'\left( h_i^t(\boldsymbol{x}_b) \right) G_{ab} \,, \tag{42}$$

there is

$$\begin{aligned}
\frac{1}{m} \sum_{i=1}^m \left| \dot{h}_i^t(\boldsymbol{x}_a) \right| &\leq \hat{c} \frac{1}{m} \sum_{i=1}^m \left| \sum_{b=1}^n \left( f^t(\boldsymbol{x}_b) - y_b \right) \sigma'\left( h_i^t(\boldsymbol{x}_b) \right) G_{ab} \right| \\
&\leq \hat{c} \left( \frac{1}{m} \sum_{i=1}^m \left| \sum_{b=1}^n \left( f^t(\boldsymbol{x}_b) - y_b \right) \sigma'\left( h_i^t(\boldsymbol{x}_b) \right) G_{ab} \right|^2 \right)^{\frac{1}{2}} \\
&\leq \hat{c} \left( \frac{1}{m} \sum_{i=1}^m \sum_{b=1}^n \left| \left( f^t(\boldsymbol{x}_b) - y_b \right) \sigma'\left( h_i^t(\boldsymbol{x}_b) \right) G_{ab} \right|^2 \right)^{\frac{1}{2}} \\
&\leq \hat{c} \lambda_{\max}(G) \left( \frac{1}{m} \sum_{i=1}^m \sum_{b=1}^n \left( f^t(\boldsymbol{x}_b) - y_b \right)^2 \left( \sigma'\left( h_i^t(\boldsymbol{x}_b) \right) \right)^2 \right)^{\frac{1}{2}} \\
&\leq \hat{c} \lambda_{\max}(G) \left( \frac{\left| \dot{\mathcal{L}}^t \right|}{(\hat{c})^2 \lambda_{\min}(G)} \right)^{\frac{1}{2}} \\
&\leq \lambda_{\max}(G) \left( \lambda_{\min}(G) \right)^{-\frac{1}{2}} \left| \dot{\mathcal{L}}^t \right|^{\frac{1}{2}}
\end{aligned} \tag{43}$$

Therefore,

$$\begin{aligned}
\frac{1}{m} \sum_{i=1}^m \left| h_i^t(\boldsymbol{x}_a) - h_i^0(\boldsymbol{x}_a) \right| &\leq \int_0^t \frac{1}{m} \sum_{i=1}^m \left| \dot{h}_i^s(\boldsymbol{x}_a) \right| ds \\
&\leq \lambda_{\max}(G) \left( \lambda_{\min}(G) \right)^{-\frac{1}{2}} \int_0^t \left| \dot{\mathcal{L}}^s \right|^{\frac{1}{2}} ds
\end{aligned} \tag{44}$$

Since $\forall \xi \in \mathbb{R}$, there is

$$\xi \cdot \mathbb{1}_{|h_i^t(\boldsymbol{x}_a) - h_i^0(\boldsymbol{x}_a)| \geq \xi} \leq \left| h_i^t(\boldsymbol{x}_a) - h_i^0(\boldsymbol{x}_a) \right| \,, \tag{45}$$

we derive that, $\forall a \in [n]$,

$$
\begin{aligned}
\frac{1}{m} \sum_{i=1}^{m} \mathbb{1}_{|h_i^t(\boldsymbol{x}_a) - h_i^0(\boldsymbol{x}_a)| \geq \xi} &\leq \xi^{-1} \frac{1}{m} \sum_{i=1}^{m} \left| h_i^t(\boldsymbol{x}_a) - h_i^0(\boldsymbol{x}_a) \right| \\
&\leq C_1 \int_0^t \left| \dot{\mathcal{L}}^s \right|^{\frac{1}{2}} ds ,
\end{aligned}
\tag{46}
$$

where we set $C_1 = \lambda_{\max}(G) \left( \lambda_{\min}(G) \right)^{-\frac{1}{2}} \xi^{-1} > 0$ for simplicity. As a consequence,

$$
\max_{a \in [n]} \left\{ \frac{1}{m} \sum_{i=1}^{m} \mathbb{1}_{|h_i^t(\boldsymbol{x}_a) - h_i^0(\boldsymbol{x}_a)| \geq \xi} \right\} \leq C_1 \int_0^t \left| \dot{\mathcal{L}}^s \right|^{\frac{1}{2}} ds
\tag{47}
$$

Define

$$
\tilde{\eta}^t = \min_{a \in [n]} \left\{ \frac{1}{m} \sum_{i=1}^{m} \mathbb{1}_{h_i^0(\boldsymbol{x}_a) \in I_0} \right\} - C_1 \int_0^t \left| \dot{\mathcal{L}}^s \right|^{\frac{1}{2}} ds
\tag{48}
$$

Note that at $t = 0$, there is $\tilde{\eta}^t = \min_{a \in [n]} \left\{ \frac{1}{m} \sum_{i=1}^{m} \mathbb{1}_{h_i^0(\boldsymbol{x}_a) \in I_0} \right\}$. Then, on one hand, we know from (40) and (47) that $\forall t \geq 0$,

$$
\eta^t \geq \min_{a \in [n]} \left\{ \frac{1}{m} \sum_{i=1}^{m} \mathbb{1}_{h_i^0(\boldsymbol{x}_a) \in I_0} \right\} - \max_{a \in [n]} \left\{ \frac{1}{m} \sum_{i=1}^{m} \mathbb{1}_{|h_i^t(\boldsymbol{x}_a) - h_i^0(\boldsymbol{x}_a)| \geq \xi} \right\} \geq \tilde{\eta}^t ,
\tag{49}
$$

Hence, (41) implies that

$$
\begin{aligned}
\dot{\mathcal{L}}^t &\leq -2\hat{c}^2 \lambda_{\min}(G) \left( K_{\sigma'} \right)^2 \mathcal{L}^t \eta^t \\
&\leq -2\hat{c}^2 \lambda_{\min}(G) \left( K_{\sigma'} \right)^2 \mathcal{L}^t \tilde{\eta}^t
\end{aligned}
\tag{50}
$$

On the other hand, by the definition of $\tilde{\eta}^t$,

$$
\begin{aligned}
\dot{\tilde{\eta}}^t &= -C_1 \left| \dot{\mathcal{L}}^t \right|^{\frac{1}{2}} \\
&\geq -C_1 \left| \dot{\mathcal{L}}^t \right| \cdot \left| \dot{\mathcal{L}}^t \right|^{-\frac{1}{2}} \\
&\geq C_1 \dot{\mathcal{L}}^t \left( 2\hat{c}^2 \lambda_{\min}(G) \left( K_{\sigma'} \right)^2 \mathcal{L}^t \tilde{\eta}^t \right)^{-\frac{1}{2}} \\
&\geq C_2 (\hat{c})^{-1} \dot{\mathcal{L}}^t \left( \mathcal{L}^t \right)^{-\frac{1}{2}} \left( \tilde{\eta}^t \right)^{-\frac{1}{2}} ,
\end{aligned}
\tag{51}
$$

where we set $C_2 = 2^{-\frac{1}{2}} C_1 \left( \lambda_{\min}(G) \right)^{-\frac{1}{2}} \left( K_{\sigma'} \right)^{-1} = \frac{\lambda_{\max}(G)}{\sqrt{2} \xi K_{\sigma'} \lambda_{\min}(G)} \leq \frac{n G_{\max}}{\sqrt{2} \xi K_{\sigma'} \lambda_{\min}(G)}$ for simplicity. Therefore, when $\eta^t > 0$,

$$
\frac{d}{dt} \left( \frac{2}{3} \left( \tilde{\eta}^t \right)^{\frac{3}{2}} \right) = \left( \tilde{\eta}^t \right)^{\frac{1}{2}} \dot{\tilde{\eta}}^t \geq C_2 (\hat{c})^{-1} \dot{\mathcal{L}}^t \left( \mathcal{L}^t \right)^{-\frac{1}{2}} \geq C_2 (\hat{c})^{-1} \frac{d}{dt} \left( 2\mathcal{L}^t \right)^{\frac{1}{2}} ,
\tag{52}
$$

which implies that

$$
\frac{2}{3} \left( \tilde{\eta}^t \right)^{\frac{3}{2}} - \frac{2}{3} \left( \tilde{\eta}^0 \right)^{\frac{3}{2}} \geq C_2(\hat{c})^{-1} \left( \left( 2\mathcal{L}^t \right)^{\frac{1}{2}} - \left( 2\mathcal{L}^0 \right)^{\frac{1}{2}} \right) \geq -C_2(\hat{c})^{-1} \left( 2\mathcal{L}^0 \right)^{\frac{1}{2}}
\tag{53}
$$

and so $\forall t \geq 0$,

$$
\frac{2}{3} \left( \eta^t \right)^{\frac{3}{2}} \geq \frac{2}{3} \left( \tilde{\eta}^t \right)^{\frac{3}{2}} \geq \frac{2}{3} \left( \tilde{\eta}^0 \right)^{\frac{3}{2}} - C_2(\hat{c})^{-1} \left( 2\mathcal{L}^0 \right)^{\frac{1}{2}}
\tag{54}
$$

## E PROOF OF THEOREM 1

To apply Lemma 1, we need two additional lemmas, which we will prove in Appendix E.1 and E.2. The first one guarantees that the loss value at initialization, $\mathcal{L}^0$, is upper-bounded with high probability:

**Lemma 2.** $\forall \delta > 0$, if $m \geq \Omega\left(\hat{c}^2 \log\left(n\delta^{-1}\right) G_{\max}/\|\boldsymbol{y}\|^2\right)$, then with probability at least $1 - \delta$, there is

$$\mathcal{L}^0 \leq \|\boldsymbol{y}\|^2 \tag{55}$$

The second one proves that $\tilde{\eta}^0$ is lower-bounded with high probability, which heuristically says that there is indeed a nontrivial proportion of neurons in the central part of the active region of $\sigma$, for every $a \in [n]$:

**Lemma 3.** $\forall \delta > 0$, if $m \geq \frac{\log(n\delta^{-1})}{2(K(I,G_{\min},G_{\max}))^2}$, then with probability at least $1 - \delta$, there is

$$\eta^0 > K(I, G_{\min}, G_{\max}) , \tag{56}$$

where $K(I, \lambda_1, \lambda_2) = \frac{1}{6\sqrt{2\pi}\lambda_2}(I_r - I_l) \exp\left\{-\frac{\max\{|I_l|,|I_r|\}}{(\lambda_1)^2}\right\}$ is a positive number that depends on $I$, $\lambda_1$ and $\lambda_2$.

With these two lemmas, we deduce that $\forall \delta > 0$, if $m \geq \Omega\left((1 + \hat{c}^2/\|\boldsymbol{y}\|^2)\log\left(n\delta^{-1}\right)\right)$, then with probability at least $\delta$, there is $\forall t \geq 0$,

$$\frac{2}{3}\left(\eta^t\right)^{\frac{3}{2}} \geq \frac{2}{3}\left(K(I, G_{\min}, G_{\max})\right)^{\frac{3}{2}} - \sqrt{2}C_2(\hat{c})^{-1}\|\boldsymbol{y}\| , \tag{57}$$

where $K(I, G_{\min}, G_{\max})$ is defined as in Lemma 3. Therefore, if our choice of $\hat{c}$ satisfies

$$\hat{c} \geq \frac{3\sqrt{2}C_2\|\boldsymbol{y}\|}{\left(K(I, G_{\min}, G_{\max})\right)^{\frac{3}{2}}} , \tag{58}$$

then there is $\forall t \geq 0$,

$$\eta^t \geq 2^{-\frac{2}{3}} K(I, G_{\min}, G_{\max}) > 0 , \tag{59}$$

in which case (50) gives

$$\dot{\mathcal{L}}^t \leq -2^{\frac{1}{3}}\hat{c}^2\lambda_{\min}(G)\left(K_{\sigma'}\right)^2 \mathcal{L}^t K(I, G_{\min}, G_{\max}) , \tag{60}$$

which will allow us to finally conclude that

$$\mathcal{L}^t \leq \mathcal{L}^0 \exp\left\{-2^{\frac{1}{3}}\lambda_{\min}(G)\left(K_{\sigma'}\right)^2 K(I, G_{\min}, G_{\max})\hat{c}^2 t\right\} \tag{61}$$

Note that (60) establishes a PL condition. Several other convergence analyses of NNs have also relied on variants of the PL condition [25, 44, 74].

### E.1 PROOF OF LEMMA 2

*Proof.* Since at initialization, $\{c_i\}_{i\in[m]}$ and $\{W_{ij}^0\}_{i\in[m],j\in[D]}$ are both sampled i.i.d. and $\{c_i\}_{i\in[m]}$ has mean zero, we know that $\forall a \in [n]$, $f^0(\boldsymbol{x}_a) = \frac{1}{m}\sum_{i=1}^m c_i\sigma\left(h_i^t(\boldsymbol{x}_a)\right)$ is the sample mean of i.i.d. random variables with zero-mean. Moreover, since $\{W_{ij}^0\}_{i\in[m],j\in[D]}$ is sampled from $\mathcal{N}(0, 1)$, we know that $\forall i \in [m]$, the random variable $c_i\sigma\left(h_i^t(\boldsymbol{x}_a)\right)$ is sub-Gaussian [68], with sub-Gaussian norm

$$
\begin{aligned}
\|c_i\sigma\left(h_i^t(\boldsymbol{x}_a)\right)\|_{\psi_2} &\leq \hat{c}\|\sigma\left(h_i^t(\boldsymbol{x}_a)\right)\|_{\psi_2} \\
&\leq \hat{c}L_\sigma(G_{aa})^{\frac{1}{2}}M_{SG} \\
&\leq \hat{c}L_\sigma(G_{\max})^{\frac{1}{2}}M_{SG} ,
\end{aligned} \tag{62}
$$

where $M_{SG} > 0$ is some absolute constant. Thus, by Hoeffding's inequality [68], $\forall a \in [n]$, $\forall r > 0$,

$$
\begin{aligned}
\mathbb{P}\left(\left|f^0(\boldsymbol{x}_a)\right| \geq u\right) &= \mathbb{P}\left(\left|\frac{1}{m}\sum_{i=1}^m c_i\sigma\left(h_i^t(\boldsymbol{x}_a)\right)\right| \geq u\right) \\
&\leq 2\exp\left\{-\frac{Ku^2m}{\|c_i\sigma\left(h_i^t(\boldsymbol{x}_a)\right)\|_{\psi_2}^2}\right\} \\
&\leq 2\exp\left\{-\frac{Ku^2m}{\hat{c}^2(L_\sigma)^2 G_{\max}(M_{SG})^2}\right\} ,
\end{aligned} \tag{63}
$$

where $K$ is some absolute constant. Hence, by union bound,

$$
\mathbb{P}\left(\sum_{a=1}^{n}\left|f^{0}(\boldsymbol{x}_{a})\right|^{2}\geq\|\boldsymbol{y}\|^{2}\right)\leq\sum_{a=1}^{n}\mathbb{P}\left(\left|f^{0}(\boldsymbol{x}_{a})\right|\geq\|\boldsymbol{y}\|\right)
$$

$$
\leq 2n\exp\left\{-\frac{K\|\boldsymbol{y}\|^{2}m}{\hat{c}^{2}(L_{\sigma})^{2}G_{\max}(M_{SG})^{2}}\right\}
\tag{64}
$$

Thus, $\forall \delta > 0$, if

$$
m\geq\hat{c}^{2}(L_{\sigma})^{2}G_{\max}K^{-1}(M_{SG})^{2}\|\boldsymbol{y}\|^{-2}\log\left(\frac{2n}{\delta}\right)
\tag{65}
$$

then with probability at least $1 - \delta$, there is

$$
\begin{aligned}
\mathcal{L}^{0} =&\frac{1}{2}\sum_{a=1}^{n}\left|f^{0}(\boldsymbol{x}_{a})-y_{a}\right|^{2}\\
\leq&\frac{1}{2}\sum_{a=1}^{n}\left|f^{0}(\boldsymbol{x}_{a})\right|^{2}+\frac{1}{2}\|\boldsymbol{y}\|^{2}\\
\leq&\|\boldsymbol{y}\|^{2}
\end{aligned}
\tag{66}
$$

$\square$

### E.2   PROOF OF LEMMA 3

Since each $W_{ij}^{0}$ are sampled i.i.d. from $\mathcal{N}(0,1)$, we know that $\forall a \in [n]$, independently for each $i \in [m]$, $h_{i}^{0}(\boldsymbol{x}_{a})$ follows a Gaussian distribution with mean 0 and variance $G_{aa}$. Therefore,

$$
\sum_{i=1}^{m}\mathbb{1}_{h_{i}^{0}(\boldsymbol{x}^{a})\notin I_{0}}\sim\text{Binomial}\left(m,1-\pi\left(I_{0};G_{aa}\right)\right) ,
\tag{67}
$$

Hence, by Hoeffding's inequality, $\forall a \in [n], \forall r > 0$,

$$
\mathbb{P}\left(\frac{1}{m}\sum_{i=1}^{m}\mathbb{1}_{h_{i}^{0}(x_{a})\notin I_{0}}\geq 1-\pi(I_{0};G_{aa})+r\right)\leq\exp\left\{-2mr^{2}\right\}
\tag{68}
$$

$\forall a \in [n]$, choosing $r = \frac{1}{2}\pi\left(I_{0};G_{aa}\right)$, we then get

$$
\begin{aligned}
\mathbb{P}\left(\frac{1}{m}\sum_{i=1}^{m}\mathbb{1}_{h_{i}^{0}(x_{a})\in I_{0}}\leq\frac{1}{2}\pi\left(I_{0};G_{aa}\right)\right) =&\mathbb{P}\left(\frac{1}{m}\sum_{i=1}^{m}\mathbb{1}_{h_{i}^{0}(x_{a})\notin I_{0}}\geq 1-\frac{1}{2}\pi\left(I_{0};G_{aa}\right)\right)\\
\leq&\exp\left\{-\frac{1}{2}m\left(\pi\left(I_{0};G_{aa}\right)\right)^{2}\right\}\\
\leq&\exp\left\{-\frac{1}{2}m\left(\min_{b\in[n]}\left\{\pi\left(I_{0};G_{bb}\right)\right\}\right)^{2}\right\}
\end{aligned}
\tag{69}
$$

and so by union bound

$$
\begin{aligned}
\mathbb{P}\left(\tilde{\eta}^{0}\leq\frac{1}{2}\min_{b\in[n]}\left\{\pi\left(I_{0};G_{bb}\right)\right\}\right) =&\mathbb{P}\left(\min_{a\in[n]}\left\{\frac{1}{m}\sum_{i=1}^{m}\mathbb{1}_{h_{i}^{0}(x_{a})\in I_{0}}\right\}<\frac{1}{2}\min_{b\in[n]}\left\{\pi\left(I_{0};G_{bb}\right)\right\}\right)\\
\leq&\sum_{a=1}^{n}\mathbb{P}\left(\frac{1}{m}\sum_{i=1}^{m}\mathbb{1}_{h_{i}^{0}(x_{a})\in I_{0}}<\frac{1}{2}\min_{b\in[n]}\left\{\pi\left(I_{0};G_{bb}\right)\right\}\right)\\
\leq&\sum_{a=1}^{n}\mathbb{P}\left(\frac{1}{m}\sum_{i=1}^{m}\mathbb{1}_{h_{i}^{0}(x_{a})\in I_{0}}<\frac{1}{2}\pi\left(I_{0};G_{aa}\right)\right)\\
\leq&n\exp\left\{-\frac{1}{2}m\left(\min_{b\in[n]}\left\{\pi\left(I_{0};G_{bb}\right)\right\}\right)^{2}\right\}
\end{aligned}
\tag{70}
$$

Since $\forall b \in [n]$, there is $G_{\min} \leq G_{bb} \leq G_{\max}$,

$$
\begin{aligned}
\pi\left(I_0; G_{bb}\right) =& \frac{1}{\sqrt{2\pi G_{bb}}} \int_{I_l+\xi}^{I_r-\xi} e^{-\frac{u^2}{G_{bb}}} du \\
\geq& \frac{1}{\sqrt{2\pi G_{bb}}} (I_r - I_l - 2\xi) \inf_{I_l+\xi \leq u \leq I_r-\xi} \exp\left\{-\frac{u^2}{G_{bb}}\right\} \\
\geq& \frac{1}{\sqrt{2\pi G_{\max}}} (I_r - I_l - 2\xi) \exp\left\{-\frac{\max\{|I_l|, |I_r|\}}{(G_{\min})^2}\right\} \\
\geq& \frac{1}{3\sqrt{2\pi G_{\max}}} (I_r - I_l) \exp\left\{-\frac{\max\{|I_l|, |I_r|\}}{(G_{\min})^2}\right\}
\end{aligned}
\tag{71}
$$

Letting $K(I, \lambda_1, \lambda_2) = \frac{1}{6\sqrt{2\pi}\lambda_2}(I_r - I_l) \exp\left\{-\frac{\max\{|I_l|, |I_r|\}}{(\lambda_1)^2}\right\} > 0$, we can then write

$$
\begin{aligned}
\mathbb{P}\left(\tilde{\eta}^0 \leq K(I, G_{\min}, G_{\max})\right) \leq& \mathbb{P}\left(\tilde{\eta}_0 \leq \frac{1}{2} \min_{b \in [n]} \{\pi\left(I_0; G_{bb}\right)\}\right) \\
\leq& n \exp\left\{-2m\left(K(I, G_{\min}, G_{\max})\right)^2\right\}
\end{aligned}
\tag{72}
$$

Thus, $\forall \delta > 0$, if $m \geq \frac{\log(n\delta^{-1})}{2(K(I, G_{\min}, G_{\max}))^2}$, then with probability at least $1 - \delta$, it holds that $\tilde{\eta}^0 > K(I, G_{\min}, G_{\max}) > 0$.

## F  PROOF OF THEOREM 2

By Condition 1, we know that $\forall \delta > 0$, if $D \geq D_{\min}(\frac{1}{2}\delta, \frac{1}{2}\lambda_{\min}(\bar{G}))$, then with probability at least $1 - \frac{1}{2}\delta$, there is $\|G - \bar{G}\|_2 \leq \frac{1}{2}\lambda_{\min}(\bar{G})$, and hence $\lambda_{\min}(G) \geq \frac{1}{2}\lambda_{\min}(\bar{G})$, $G_{\min} \geq \frac{1}{2}\bar{G}_{\min}$, $\lambda_{\max}(G) \leq \lambda_{\max}(\bar{G}) + \frac{1}{2}\lambda_{\min}(\bar{G}) \leq 2\lambda_{\max}(\bar{G})$, and $G_{\max} \leq 2\bar{G}_{\max}$. We then perform the following analysis conditioned on the event that $\|G - \bar{G}\|_2 \leq \frac{1}{2}\lambda_{\min}(\bar{G})$.

Since the sampling of $\{c_i\}_{i \in [m]}$ and $\{W_{ij}^0\}_{i \in [m], j \in [D]}$ is independent from the realization of $G$, we know from Lemma 3 that if $m \geq \frac{\log(4n\delta^{-1})}{2\left(K(I, \frac{1}{2}\lambda_{\min}(\bar{G}), 2\lambda_{\max}(\bar{G}))\right)^2} \geq \frac{\log(4n\delta^{-1})}{2(K(I, G_{\min}, G_{\max}))^2}$, then with probability at least $1 - \frac{1}{4}\delta$, there is

$$
\tilde{\eta}^0 > K(I, G_{\min}, G_{\max}) \geq K(I, \frac{1}{2}\bar{G}_{\min}, 2\bar{G}_{\max})
\tag{73}
$$

From Lemma 2, we also know that if $m \geq \Omega\left(\hat{c}^2 \log\left(n\delta^{-1}\right)\lambda_{\max}(\bar{G})/\|\boldsymbol{y}\|^2\right) \geq \Omega\left(\hat{c}^2 \log\left(n\delta^{-1}\right)\lambda_{\max}(G)/\|\boldsymbol{y}\|^2\right)$, then with probability at least $1 - \frac{1}{4}\delta$, there is $\mathcal{L}^0 \leq \|\boldsymbol{y}\|^2$. Therefore, in total, we know that with probability at least $1 - \delta$, the following conditions all hold:

$$
\|G - \bar{G}\|_2 \leq \frac{1}{2}\lambda_{\min}(\bar{G}) \,,
\tag{74}
$$

$$
\tilde{\eta}^0 \geq K(I, \frac{1}{2}\bar{G}_{\min}, 2\bar{G}_{\max}) \,,
\tag{75}
$$

$$
\mathcal{L}^0 \leq \|\boldsymbol{y}\|^2 \,,
\tag{76}
$$

in which case, by applying Lemma 1 with $G = G^{(1)}$, we get

$$
\left(\eta^t\right)^{\frac{3}{2}} \geq \left(\tilde{\eta}^0\right)^{\frac{3}{2}} - K_1(\lambda_{\min}(G), \lambda_{\max}(G))(\hat{c})^{-1}\left(\left(\mathcal{L}^0\right)^{\frac{1}{2}} - \left(\mathcal{L}^t\right)^{\frac{1}{2}}\right) \,,
\tag{77}
$$

where $K_1(\lambda_1, \lambda_2) = \frac{9}{2}\lambda_1^{-1}\lambda_2 K_{\sigma'}^{-1}(I_r - I_l) > 0$. Thus, by the definition of $K_1(\cdot, \cdot)$, we know that

$$
K_1(\lambda_{\min}(G), \lambda_{\max}(G)) \leq 4K_1(\lambda_{\min}(\bar{G}), \lambda_{\max}(\bar{G})) \,,
\tag{78}
$$

and so $\forall t \geq 0$,

$$
\begin{aligned}
\left(\eta^t\right)^{\frac{3}{2}} \geq& \left(\tilde{\eta}^0\right)^{\frac{3}{2}} - 4K_1(\lambda_{\min}(\bar{G}), \lambda_{\max}(\bar{G}))(\hat{c})^{-1}\left(\left(\mathcal{L}^0\right)^{\frac{1}{2}} - \left(\mathcal{L}^t\right)^{\frac{1}{2}}\right) \\
\geq& \left(K(I, \frac{1}{2}\bar{G}_{\min}, 2\bar{G}_{\max})\right)^{\frac{3}{2}} - 4K_1(\lambda_{\min}(\bar{G}), \lambda_{\max}(\bar{G}))(\hat{c})^{-1}\|\boldsymbol{y}\|
\end{aligned}
\tag{79}
$$

Therefore, if our choice of $\hat{c}$ satisfies

$$\hat{c} \geq \frac{8K_1(\lambda_{\min}\left(\bar{G}\right), \lambda_{\max}\left(\bar{G}\right))\|\boldsymbol{y}\|}{\left(K(I, \frac{1}{2}\bar{G}_{\min}, 2\bar{G}_{\max})\right)^{\frac{3}{2}}} \tag{80}$$

then there is $\forall t \geq 0$,

$$\eta^t \geq 2^{-\frac{2}{3}} K(I, \frac{1}{2}\bar{G}_{\min}, 2\bar{G}_{\max}) \tag{81}$$

Hence, (50) implies that $\forall t \geq 0$,

$$\begin{aligned}
\dot{\mathcal{L}}^t \leq &-2\hat{c}^2\lambda_{\min}(G)(K_{\sigma'})^2 \mathcal{L}^t 2^{-\frac{2}{3}} K(I, \frac{1}{2}\bar{G}_{\min}, 2\bar{G}_{\max}) \\
\leq &-2^{-\frac{2}{3}}\hat{c}^2\lambda_{\min}(\bar{G})(K_{\sigma'})^2 \mathcal{L}^t K(I, \frac{1}{2}\bar{G}_{\min}, 2\bar{G}_{\max})
\end{aligned} \tag{82}$$

and therefore

$$\mathcal{L}^t \leq \mathcal{L}^0 \exp\left\{-2^{-\frac{2}{3}}\hat{c}^2\lambda_{\min}(\bar{G})(K_{\sigma'})^2 K(I, \frac{1}{2}\bar{G}_{\min}, 2\bar{G}_{\max})\right\} \tag{83}$$

## G   PROOF OF THEOREM 3

In view of Theorem 2, it is sufficient to verify that Condition 1 holds for $D_{\min}(\delta, u) = \Omega\left(n^2 u^{-2} \log(n\delta^{-1})\right)$, which is given by the following lemma:

**Lemma 4.** $\forall \delta \geq 0$, if $D \geq \Omega\left(n^2 u^{-2} \log(n\delta^{-1})\right)$, then with probability at least $1 - \delta$,

$$\|G^{(1)} - \bar{G}^{(1)}\|_2 \leq u \tag{84}$$

### G.1   PROOF OF LEMMA 4

Let $Z$ be a random vector on $\mathbb{R}^d$ with law given by $\pi_{\boldsymbol{z}}$, and then we can write $\bar{G}_{ab}^{(1)} = \mathbb{E}\left[\sigma(\boldsymbol{x}_a^\intercal Z)\sigma(\boldsymbol{x}_b^\intercal Z)\right]$ for $a, b \in [n]$. By assumption, $Z$ is sub-gaussian with sub-gaussian norm

$$\|Z\|_{\psi_2} := \sup_{x \in S^{d-1}} \|x^\intercal Z\|_{\psi_2} < \infty \tag{85}$$

Thus, $\forall a \in [n]$, we have

$$\|\sigma(x_a^\intercal Z)\|_{\psi_2} \leq L_\sigma \|x^\intercal Z\|_{\psi_2} \leq L_\sigma \|Z\|_{\psi_2} \tag{86}$$

Hence, by Lemma 2.7.7 in [68], we know that $\forall a, b \in [n]$, $\sigma(x_a^\intercal Z)\sigma(x_b^\intercal Z)$ is a sub-exponential random variable with sub-exponential norm

$$\|\sigma(x_a^\intercal Z)\sigma(x_b^\intercal Z)\|_{\psi_1} \leq \|\sigma(x_a^\intercal Z)\|_{\psi_2}\|\sigma(x_b^\intercal Z)\|_{\psi_2} \leq (L_\sigma)^2\|Z\|_{\psi_2}^2 \tag{87}$$

Then, by Bernstein's inequality (Theorem 2.8.1 in [68]), since each $\boldsymbol{z}_j$ is sampled i.i.d. from $\pi_{\boldsymbol{z}}$, we have that $\forall a, b \in [n]$ and $\forall u > 0$,

$$\begin{aligned}
\mathbb{P}\left(\left|G_{ab}^{(1)} - \bar{G}_{ab}^{(1)}\right| \geq u\right) = &\mathbb{P}\left(\left|\frac{1}{D}\sum_{j=1}^{D}\sigma(x_a^\intercal z^j)\sigma(x_b^\intercal z^j) - \mathbb{E}\left[\sigma(x_a^\intercal Z)\sigma(x_b^\intercal Z)\right]\right| \geq u\right) \\
\leq &2\exp\left\{-K\min\left\{\frac{u^2 D}{\|\sigma(x_a^\intercal Z)\sigma(x_b^\intercal Z)\|_{\psi_1}^2}, \frac{uD}{\|\sigma(x_a^\intercal Z)\sigma(x_b^\intercal Z)\|_{\psi_1}}\right\}\right\} \\
\leq &2\exp\left\{-K\min\left\{\frac{u^2 D}{(L_\sigma)^4\|Z\|_{\psi_2}^4}, \frac{uD}{(L_\sigma)^2\|Z\|_{\psi_2}^2}\right\}\right\},
\end{aligned} \tag{88}$$

where $K > 0$ is some absolute constant. In other words, for any $\delta' > 0$, if

$$D \geq \max\left\{\frac{(L_\sigma)^4\|Z\|_{\psi_2}^4\log(2(\delta')^{-1})}{u^2 K}, \frac{(L_\sigma)^2\|Z\|_{\psi_2}^2\log(2(\delta')^{-1})}{uK}\right\}, \tag{89}$$

then we have $\left| G_{ab}^{(1)} - \bar{G}_{ab}^{(1)} \right| \geq u$ with probability at least $1 - \delta$. If we choose $u = \frac{u'}{n}$ and $\delta' = \frac{\delta}{n^2}$, then we get, if

$$D \geq \max \left\{ \frac{n^2 (L_\sigma)^4 \|Z\|_{\psi_2}^4 \log(2n^2 \delta^{-1})}{K(u')^2}, \frac{n(L_\sigma)^2 \|Z\|_{\psi_2}^2 \log(2n^2 \delta^{-1})}{Ku'} \right\} , \tag{90}$$

then

$$\begin{aligned} \mathbb{P}\left( \|G^{(1)} - \bar{G}^{(1)}\|_F^2 \geq (u')^2 \right) &\leq \sum_{a,b=1}^{n} \mathbb{P}\left( \left| G_{ab}^{(1)} - \bar{G}_{ab}^{(1)} \right| \geq \frac{u'}{n} \right) \\ &\leq n^2 \frac{\delta}{n^2} \\ &\leq \delta \end{aligned} \tag{91}$$

Hence, with probability at least $1 - \delta$, we have

$$\|G^{(1)} - \bar{G}^{(1)}\|_2^2 \leq \|G^{(1)} - \bar{G}^{(1)}\|_F^2 \leq \left(u'\right)^2 , \tag{92}$$

## H  GENERALIZATION TO DEEPER MODELS

By setting $\Phi$ to be the activations of the second-to-last hidden-layer of a multi-layer NN, we can obtain generalizations of the P-3L NN to deeper architectures. For example, in the feed-forward case, we can obtain the following *partially-trained L-layer NN*:

$$f(\boldsymbol{x}) = \frac{1}{m} \sum_{i=1}^{m} c_i \sigma\left(h_i^{(L-1)}(\boldsymbol{x})\right) ,$$

$$\forall i \in [m] \quad : \qquad h_i^{(L-1)}(\boldsymbol{x}) = \frac{1}{\sqrt{D}} \sum_{j=1}^{D} W_{ij} \sigma\left(h_j^{(L-2)}(\boldsymbol{x})\right) ,$$

$$\forall l \in [L-3], \forall i \in [D] \quad : \qquad h_i^{(l+1)}(\boldsymbol{x}) = \frac{1}{\sqrt{D}} \sum_{j=1}^{D} \bar{W}_{ij}^{(l)} \sigma\left(h_j^{(l)}(\boldsymbol{x})\right) ,$$

$$\forall j \in [D] \quad : \qquad h_j^{(1)}(\boldsymbol{x}) = \frac{1}{\sqrt{d}} \boldsymbol{z}_j^{\mathsf{T}} \boldsymbol{x} ,$$

where $\bar{W}^{(1)}, ..., \bar{W}^{(L-3)} \in \mathbb{R}^{D \times D}$ and $\boldsymbol{z}_1, ..., \boldsymbol{z}_D \in \mathbb{R}^d$ are sampled randomly and fixed. This model can be written in the form of (4) and (5) with $\phi_j(\boldsymbol{x}) = \sigma\left(h_j^{(L-2)}(\boldsymbol{x})\right)$. The corresponding Gram matrix is recursively defined and also appears in the NTK analysis [18]. In particular, the results in [18] imply that if $\sigma$ is analytic and not a polynomial, then Condition 1 holds, and hence similar global convergence results can be obtained as corollaries of Theorem 2.

## I  FURTHER DETAILS OF THE NUMERICAL EXPERIMENTS

In our models, $\left\{c_i\right\}_{i \in [m]}$ is sampled i.i.d. from the Rademacher distribution $\mu_c = \frac{1}{2}\boldsymbol{\delta}_1 + \frac{1}{2}\boldsymbol{\delta}_{-1}$, $\left\{\boldsymbol{z}_j\right\}_{j \in [D]}$ is sampled i.i.d. from $\mathcal{N}(0, I_d)$, and $\left\{W_{ij}\right\}_{i \in [m], j \in [D]}$ is initialized by sampling i.i.d. from $\mathcal{N}(0, 1)$. In the model under NTK scaling, we additionally symmetrize the model at initialization according to the strategy used in [16] to ensure that the function value at initialization does not blow up when the width is large. We choose to train the models using $50000$ steps of (full-batch) GD with step size $\delta = 1$. When the test error is computed, we use a test set of size $500$ generated by sampling i.i.d. from the same distribution as the training set.

The experiments are run with NVIDIA GPUs (1080ti and Titan RTX).

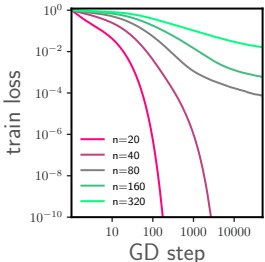

Figure 5: Training loss v.s. number of GD steps for different $n$ in Experiment 1 with $m = 4096$.

## I.1 EXPERIMENT 1

We choose $\sigma$ to be tanh. For each choice of $n$, we run the experiment with 5 different random seeds, and Figure 1 plots the evolution of the training loss during GD averaged over the 5 runs with $m = 8192$.

Figure 5 is the same as Figure 1 except for having $m = 4096$. We see that the two two plots agree well.

## I.2 EXPERIMENT 2

We choose $\sigma$ to be ReLU. For each choice of $n$ and each of the two models, we experiment with 5 different random seeds, and Figure 2 plots the test error at the 50000 GD step averaged over the 5 runs $\pm$ its standard deviation.

In Figure 6, we plot the evolution of the training loss and test error during GD for the two different models, with $m = 2048$ or $8192$ and different choices of $n$, averaged over 5 runs with different random seeds. We see in particular that the difference between the two choices of $m$ is negligible, suggesting that it is unlikely to obtain performance improvements with further over-parameterization.

## I.3 EXPERIMENT 3

We choose $\sigma$ to be ReLU and input dimension $d = 50$. We use a training set of size $n = 600$ for the results reported in Figure 4. The data set is inspired by [69]: We sample both the training and the test set i.i.d. from the distribution $(\boldsymbol{x}, y) \sim \mathcal{D}$ on $\mathbb{R}^{d+1}$, under which the joint distribution of $(x_1, x_2, y)$ is

$$\mathbb{P}(x_1 = 1, x_2 = 0, y = 1) = \frac{1}{4} \tag{93}$$

$$\mathbb{P}(x_1 = -1, x_2 = 0, y = 1) = \frac{1}{4} \tag{94}$$

$$\mathbb{P}(x_1 = 0, x_2 = 1, y = -1) = \frac{1}{4} \tag{95}$$

$$\mathbb{P}(x_1 = 0, x_2 = -1, y = -1) = \frac{1}{4} \tag{96}$$

$$\tag{97}$$

and $x_3, ..., x_d$ each follow the uniform distribution in $[-1, 1]$, independently from each other as well as $x_1, x_2$ and $y$.

Figures 7 and 8 are the same as Figure 4 except for having $n = 400$ and $800$, respectively. We see that as $n$ increases, test error improves for all three models, while our P-3L NN model remains the one achieving the lowest test error.

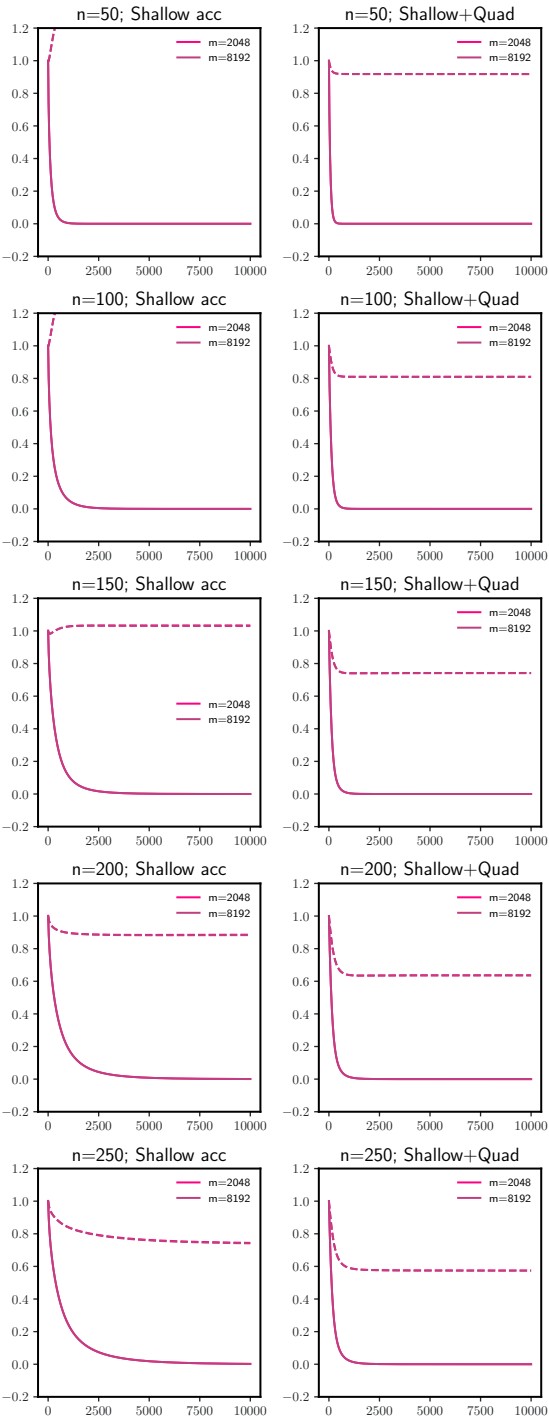

Figure 6: Test error v.s. GD steps in Experiment 2 for the two models and difference choices of $n$.

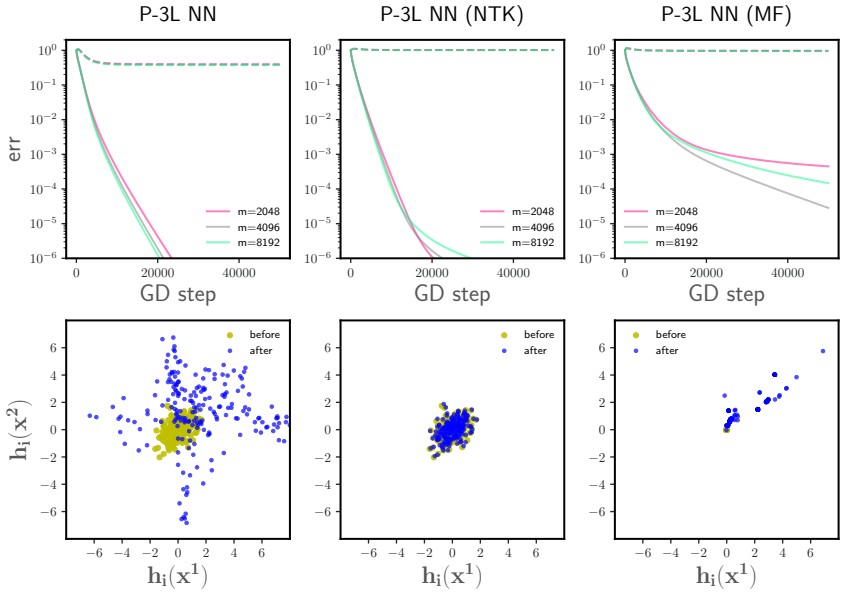

Figure 7: Same as Figure 4 except for setting $n = 400$.

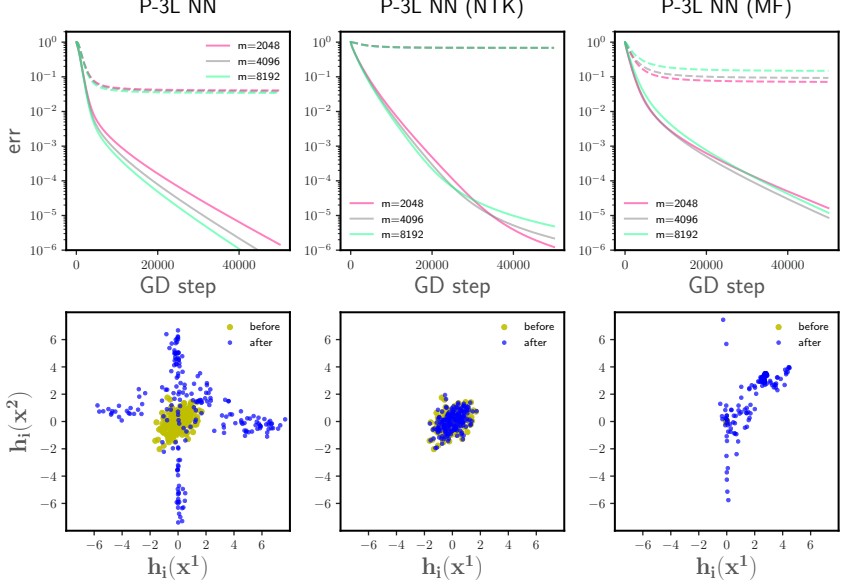

Figure 8: Same as Figure 4 except for setting $n = 800$.

