# OpenReview forum: "On feature learning in neural networks with global convergence guarantees"
_ICLR.cc/2022/Conference — ICLR 2022 Poster_

### Official Review · Reviewer_48yw · 2021-10-27

**Correctness:** 1
**Technical Novelty And Significance:** 1
**Empirical Novelty And Significance:** 1
**Recommendation:** 3
**Confidence:** 4

**Main Review:**

As far as I can tell, the setting of this paper is still limited to the lazy training regime. In other words, the essential model of fitting the $n$ data points is following linearized model:
$$
f(x;W+\Delta) \approx f(x;W) + \langle \nabla_W f(x;W), \Delta \rangle.
$$

This can be seen as follows.

- Notice that for the output of ``feature function'' $h_i(x)=\frac{1}{\sqrt{D}}\sum_{j=1}^D W_{i,j}\phi_j(x)$ changing an $O(1)$ value, $\{W_{i,j}\}$ only needs to change $O(1/\sqrt{D})$. This change shrinks as increasing $D$. Also it is indeed required that $D\geq poly(n)$ in the proof. In contrast, in Theorem 1, they only require m (the width of the output layer which is in LNN scaling) to be larger than $\log(n)$.

- On the other hand, the movement of W can roughly be estimated as follows,
  $$
  ||\Delta|| \leq \int_0^\infty ||\nabla_W L(W_t)|| d t \leq \int_0^\infty \sqrt{\lambda_\max(G) L(W_t)} dt\leq \sqrt{\lambda_\max(G)}\int_0^\infty e^{-r\lambda_\min(G) t/2}dt \leq c\sqrt{\frac{\lambda_\max(G)}{\lambda_\min(G)}} = O(poly(n)).
  $$
  Hence, on average, for each $i,j$, $W_{i,j}$ roughtly only moves $poly(n)/\sqrt{D}\ll 1$ when $D$ is sufficiently large.

Please feel free to correct me if you think I misunderstood the results.

**Summary Of The Paper:**

This paper studies the optimization of a pseudo-three-layer network. The first (output) layer is scaled by a law of large number (LNN) scaling, while the second layer is scaled by a central limit theorem (CLT)-scaling. The inputs are fixed random feature embeddings. Only the weights of the second layer are learnable. They proved that when the second layer is sufficiently over-parameterized (wrt the number of samples), GD flow converges to a zero-loss solution exponentially fast. In particular, the authors claim that, unlike the NTK regime, their setting exhibits feature learnings.






**Summary Of The Review:**

The studied setting is not too much different from previous work of non-convex optimization in the lazy regime. Therefore, I cannot recommend accepting this paper.

---

### Official Review · Reviewer_XJCF · 2021-10-27

**Correctness:** 4
**Technical Novelty And Significance:** 3
**Empirical Novelty And Significance:** 1
**Recommendation:** 8
**Confidence:** 2

**Main Review:**


Although the n <= d setting is somewhat limited, I quite liked this paper, as all previous mean field neural network analyses were quite complicated.  This one also appears novel in that there is the connection to kernels which was unexpected.  I should note I am not an expert on mean field analysis of neural networks, but to my knowledge the paper's contributions are novel and commendable.

There were a number of points of clarification I'm hoping the authors could comment on.

(1) The choice of the scaling in the network is non-standard and deserves more discussion.  The authors claim at top of pg. 4 that the scaling agrees with the scaling of Yang & Hu '20---I assume the authors are referring to the \mu P scaling in Table 1. But it seems to me that this is quite a different scaling than the one there, and also the standard mean field scalings.  The \mu P scaling requires the scaling of sqrt(D), not 1/sqrt(D), for the inner layer weights, since a_1 = -1/2.  Moreover, the learning rate scaling in \mu P is order 1, while the learning rate scaling in this paper is order m.  I understand the need for learning rate scaling of order m; this is standard in mean field and is explained well in Appendix B.

What happens when the scaling is 1/D vs. 1/sqrt(D)?  Or even unit scaling (as is standard in mean field, see MFP column in Yang & Hu '20)?  My understanding is that it only changes the scaling of lambda_min(G), and thus doesn't really affect the rates (at least in the instance of leaky relu activations).

(2) Regarding Experiment 1: Are different d chosen for the different n?  The results only hold when d <= n, so it's odd to consider d=20 for each of them but this seems to be what the authors have done.  Regarding the negative result of (65; Wojtowytsch & E '20), please give more details on what precisely the negative result is, and how your experiments relate to it.  My understanding is those results show that the risk can only decrease at rate t^{-c/d}, so that as d increases the rate should get worse.  But here you are increasing n.  It was also unclear to me why such low dimensions and sample sizes were used here.


Typos and minor comments

"feature training" on page 2, bottom of Sec 1.1

references for claim at bottom of page 2 about "weights would lease to collapse of the diversity"?

The usage of summation terms everywhere rather than vectorized forms was distracting and atypical.  If accepted, for the camera ready I would recommend re-writing everything in the form of e.g. f(x) = (1/m) c^T \sigma(Wx)

Presumably \hat c = \Theta(1) in assumption 1; if c depends on m things could go badly?

i\in [m] on top of pg.6, not i\in [n]

(30) should be \partial \mathcal L, not \partial f, presumably

"Appendix ??" on pg 17

I should note that  I did not check the details of all the proofs in the appendix.

**Summary Of The Paper:**

The authors consider the training dynamics of neural networks under a modification of the mean field parameterization in the high dimensional (d >= n) setting.  They provide an especially simple global convergence result under very mild assumptions (beyond the more restrictive d>=n one).  The proof is particularly clean and simple in the setting of leaky-relu type activations, and somewhat more technical arguments are required to get the result to hold for more general activations.  The d>=n assumption is key, as their results depend upon the minimal eigenvalue of the empirical kernel matrix defined by the inner product K(x,x') = <x, x'>, which is strictly positive when d>=n but not necessarily otherwise.  They extend the results to the feature map/embedding setting, assuming the data come from feature maps \phi(x) where \phi(x) lies in a sufficiently high dimensional space so that the kernel defined over embeddings is positive definite provided the dimension of the embedding is large enough.

**Summary Of The Review:**

The paper makes a novel contribution for convergence of NNs in the mean field regime with a clean and simple proof.

---

### Official Review · Reviewer_jNHj · 2021-10-31

**Correctness:** 3
**Technical Novelty And Significance:** 3
**Empirical Novelty And Significance:** Not applicable
**Recommendation:** 8
**Confidence:** 4

**Main Review:**

[Contributions]

Recently, the mean field setting of neural networks has become an important topic in the context of global convergence analysis of neural networks because of the presence of feature learning whereas the kernel (lazy) regime basically describes the local behavior of the dynamics of training. However, an optimization theory in the mean field regime is significantly more challenging and usually requires involved mathematical tools.

In this sense, this study makes a certain contribution in finding a simple condition. The proof can be considered as a sort of extension of the NTK-theory. Indeed, this theory basically builds upon showing the positivity of (finite-width) NTK as well as NTK-theory but does not require fixation of NTK, unlike NTK-theory. Specifically, the positivity of NTK can be reduced to the positivity of the Gram-matrix of input (random) features and $d \geq n$ by decoupling the parameter-dependent part from the NTK.

[Improvements]

- It is misleading to say that this theory covers multi-layer neural networks because the trainable layer is limited to the second-to-last layer. Therefore, the model in this paper is essentially two-layer neural networks with random feature inputs.
- The explanation of feature learning in the last paragraph of Section 2 is insufficient. In particular, the large stepsize in proportion to the width $m$ is also needed to exhibit feature learning as well as network scaling $1/m$. (See feature learning paper [67] for the detail. Although this submission considers the continuous-time dynamics, this point should be mentioned.)
- Some important references are missing. For instance, (modified) PL-condition based analysis of overparameterized neural networks is relevant to this submission in the sense that the theory essentially relies on the positivity of NTK.

    - Spencer Frei and Quanquan Gu. Proxy Convexity: A Unified Framework for the Analysis of Neural Networks Trained by Gradient Descent. NeurIPS, 2021.

    - Chaoyue Liu, Libin Zhu, and Mikhail Belkin. Loss landscapes and optimization in over-parameterized non-linear systems and neural networks. 2020.

    - Mo Zhou, Rong Ge, and Chi Jin. A Local Convergence Theory for Mildly Over-Parameterized Two-Layer Neural Network. COLT, 2021.


[Additional references]

More recently, global convergence rate analyses in the mean field regime under KL-regularization were established by several papers.

- Kaitong Hu, Zhenjie Ren, David Siska, and Lukasz Szpruch. Mean-field Langevin dynamics and energy landscape of neural networks. 2019.

- Jean-François Jabir, David Šiška, and Łukasz Szpruch. Mean-Field Neural ODEs via Relaxed Optimal Control. 2019.

- Atsushi Nitanda, Denny Wu, and Taiji Suzuki. Particledual averaging: Optimization of mean field neural networks with global convergence rate analysis. NeurIPS, 2021.

[Minor comments]

- Typo (line2, page 6): $i \in [n]$ → $i \in [m]$.
- There are typos in the definition of pseudo-L-layer NN in Section 3.2.2:
The description of $\phi_j$ seems redundant.
For $h_i^{L-1}$, the index should be $i \in [m]$.


**Summary Of The Paper:**

This paper establishes the global convergence analysis, with a linear convergence rate, of gradient flow for the neural networks in the mean field regime which possesses a feature learning aspect. The contribution of this paper is to show that the positivity of the Gram-matrix of input (random) features is sufficient for guaranteeing global convergence.

**Summary Of The Review:**

I think this paper makes a certain contribution in finding out a simple condition for the global convergence analysis in the mean-field regime. However, the quality of the manuscript could be improved.

---

### Official Review · Reviewer_zizE · 2021-11-02

**Correctness:** 4
**Technical Novelty And Significance:** 2
**Empirical Novelty And Significance:** 2
**Recommendation:** 6
**Confidence:** 3

**Main Review:**

Strength
-	Understanding the optimization of neural network beyond NTK (lazy training) is an important direction.
-	Under certain condition on the Gram matrix, it can be shown that GD converges to 0 training loss. Empirically, the neural network indeed shows the ability of feature learning.

Weakness:
-	Major concern: I was wondering if authors could clarify the convergence rate dependency on \lambda_min (G) and \lambda_max (G) in Theorem 1, Theorem 2 and Theorem 3, i.e., the relation between r,\hat{c}_min,C and \lambda_min (G), \lambda_max(G). There seems no discussion about it. It seems to me that \hat{c}_min exponentially depends on 1/\lambda_min (G) (based on Lemma 3, especially the definition of K(I,\lambda_1,\lambda_2)), which implies the width m exponentially depends on 1/\lambda_min (G). If this is true, I feel the results in this paper are less interesting, since it would require exponentially number of neurons for the global convergence.

-	For multi-layer neural network, only the second-to-last layer is trained while all other layers are random sampled and fixed. This is different from the setting in practice. It is understandable that there might be technical challenges when analyzing the case of training all layers, so I would not view this as a major limitation.

Minor comments:
-	At the end if page 3 (fixed embedding): \phi_j(x) = x_j instead of \bm{x}_j


**Summary Of The Paper:**

In this paper, the authors studied the optimization problem of shallow and deep neural network. They showed that under the non-degeneracy condition on certain Gram matrix, gradient descent (GD) can converge to 0 training loss efficiently. One important difference with the existing NTK and mean-field literatures is that a different scaling factor was used in this paper. Experiment results show that neural network with this scaling is different from NTK and mean-field scale, while it is still able to do feature learning.

**Summary Of The Review:**

Based on my above comments, I currently would not vote for accept because of the major concern above.

---

### Decision · Program_Chairs · 2022-01-20

**Decision:**

Accept (Poster)

**Comment:**

This paper studies optimization of over-parametrized neural networks in the mean-field scaling. Specifically, when the input dimension in larger than the number of training samples, the paper shows that the training loss converges to 0 at a linear rate under gradient flow. It's possible to extend the result by random feature layers to handle the case when input dimension is low. Empirically the dynamics in this paper seems to achieve better generalization performance than the NTK counterpart, but no theoretical result is known. Overall this is a solid contribution to the hard problem of analyzing the training dynamics of mean-field regime. There was some debate between reviewers on what is the definition of "feature learning" and I recommend the authors to give an explicit definition of what they mean (and potentially use a different term).